# Reconstruction of the tumor spatial microenvironment along the malignant-boundary-nonmalignant axis

Zhenzhen Xun[1,2], Xinyu Ding[1,2], Yao Zhang[3], Benyan Zhang[4], Shujing Lai[2], Duowu Zou[3], Junke Zheng[5], Guoqiang Chen [6], Bing Su [1,2], Leng Han [7] ✉ & Youqiong Ye [1,2] ✉

Although advances in spatial transcriptomics (ST) enlarge to unveil spatial landscape of tissues, it remains challenging to delineate pathology-relevant and cellular localizations, and interactions exclusive to a spatial niche (e.g., tumor boundary). Here, we develop Cottrazm, integrating ST with hematoxylin and eosin histological image, and single-cell transcriptomics to delineate the tumor boundary connecting malignant and non-malignant cell spots in tumor tissues, deconvolute cell-type composition at spatial location, and reconstruct cell type-specific gene expression profiles at sub-spot level. We validate the performance of Cottrazm along the malignant-boundary-nonmalignant spatial axis. We identify specific macrophage and fibroblast subtypes localized around tumor boundary that interacted with tumor cells to generate a structural boundary, which limits T cell infiltration and promotes immune exclusion in tumor microenvironment. In this work, Cottrazm provides an integrated tool framework to dissect the tumor spatial microenvironment and facilitates the discovery of functional biological insights, thereby identifying therapeutic targets in oncologic ST datasets.

The tumor microenvironment (TME) consists of a variety of resident or infiltrating host cells (e.g., malignant cells, immune cells, and stromal cells) and non-cellular components (e.g., secreted factors, extracellular matrix proteins), all of which have important effects on tumorigenesis, progression, and metastasis[1–4] and are associated with response to immune checkpoint blockade (ICB) therapy[3,5–8]. However, the TME varies greatly by location and tissue, including cellular composition and cell-cell interactions. The tumor boundary is a niche composed of malignant cells in the outermost circle of solid tumor and non-malignant cells that are closely adjacent in spatial architecture, bridging these distinct spatial regions. Examples of these distinct boundaries include tumor-specific keratinocytes (TSKs) residing within a fibrovascular niche at leading edges[9], multiple tumor statuses in prostate cancer[10] and infiltration of immune cells in the tumor

[1]Center for Immune-Related Diseases at Shanghai Institute of Immunology, Department of Gastroenterology, Ruijin Hospital, Shanghai Jiao Tong University School of Medicine, Shanghai, China. [2]Shanghai Institute of Immunology, State Key Laboratory of Oncogenes and Related Genes, Department of Immunology and Microbiology, Shanghai Jiao Tong University School of Medicine, Shanghai 200025, China. [3]Department of Gastroenterology, Ruijin Hospital Affiliated to Shanghai Jiao Tong University School of Medicine, Shanghai 200025, China. [4]Department of Pathology, Ruijin Hospital Affiliated to Shanghai Jiao Tong University School of Medicine, Shanghai 200025, China. [5]Hongqiao International Institute of Medicine, Shanghai Tongren Hospital, Key Laboratory of Cell Differentiation and Apoptosis of Chinese Ministry of Education, Faculty of Basic Medicine, Shanghai Jiao Tong University School of Medicine, Shanghai 200025, China. [6]State Key Laboratory of Oncogenes and Related Genes, and Research Unit of Stress and Cancer, Chinese Academy of Medical Sciences, Shanghai Cancer Institute, Renji hospital, Shanghai Jiao Tong University School of Medicine (SJTU-SM), Shanghai 200127, China. [7]Center for Epigenetics and Disease Prevention, Institute of Biosciences and Technology, Texas A&M University, Houston, TX 77030, USA. ✉e-mail: leng.han@tamu.edu; youqiong.ye@shsmu.edu.cn

boundary in liver cancer[11]. Although cellular composition, cell interactions and molecular network regulation of the tumor boundary all have profound effects on TME remodeling, the tumor boundary has been subjectively described by scientists or pathologists based on immunohistochemistry (IHC) staining data without clear criteria or consistent methods[9]. Currently, lack of experimental or bioinformatic tools to effectively delineate the tumor boundary. Tumor heterogeneity is characterized by genetic variation and copy number variations (CNVs)[12–16]; the CNV score calculated from spatial transcriptomic (ST) data for distinct spots can reflect the proportion of malignant cells and help determine core tumor regions. Exploitation of these characteristics provides an opportunity to determine the tumor boundary that contains malignant and non-malignant cells.

The advantages of ST combined with single-cell RNA sequencing (scRNA-seq) enable gene expression profiling coupled with two-dimensional spatial information directly within tissues[17,18]. Compared with clustering methods in scRNA-seq analysis, ST needs more comprehensive and integrative considerations to evaluate gene expression, spatial location, and histological information[17,19–21]. Many in situ capturing technologies, such as 10X Genomics Visium, utilized 5,000 barcoded spots with a diameter of $55–100\,\mu m$ to record mRNA positions within a $6.5 \times 6.5\,mm$ capture area. This approach is liable to include multiple homogeneous or heterogeneous cells (1–10 cells per spot) within a spot, making it challenging to distinguish cell identities in the mixed spots. Conventional bioinformatics tools for ST analysis typically consider image analysis, cell-type identification, deconvolution, spatial distribution, cell-cell communication, spatial expression patterns, the interplay of regulators in spatial location, and subcellular resolution[22]. Most tools for cell type identification in ST data are based on either cell-type mapping or cell-type deconvolution[23]. Cell-type mapping methods generally infer the most likely cell type based on gene expression or in combination with imaging data or neighboring spots[23–27] which loses the actual cell composition. Cell-type deconvolution methods generally depend on scRNA-seq data as a reference to infer the cellular composition in each spot or location[18], but do not consider the location of spots and the morphological characteristics, which may neglect the impact of spatial structure on cellular composition[18,23]. In addition, there is currently no effective method to reconstruct the expression matrix of different cell types in the same spot with high resolution, which limits research on the interaction among different cell types in the same spot and the identification of potential targets for specific cell type markers in spatial architecture. Here, we show Cottrazm, an integrated tool framework able to construct the microenvironment around the tumor boundary based on spatial transcriptomics by 10× Genomics Visium platrform. Cottrazm determines the tumor boundary connecting the malignant and non-malignant cell spots (*Cottrazm-BoundaryDefine*). ST data are adjusted based on the morphological characteristics of the samples. Then the spots of the tumor core are determined according to clustering of the morphologically adjusted expression matrix and the high CNV characteristics of the tumor. Next, a hexagonal system is used to continuously extrapolate the neighbors of tumor core spots and calculate the UMAP distance from adjacent spots to the tumor centroid. This method is able to determine whether the neighbor spot is tumor or boundary (Bdy). By integrating scRNA-seq, spatial transcriptomics, and the location of spots to generate signature score, enrichment score, and topics, Cottrazm is subsequently able to deconvolute the cellular composition of the spots (*Cottrazm-SpatialDecon*). Finally, Cottrazm reconstructed cell type-specific gene expression profiles at the sub-spot level based on spatial transcriptomics and scRNA-seq reference (*Cottrazm-SpatialRecon*). By using simulated spatial transcriptomics data, we exhibited the performance of Cottrazm by predicting the cell-type proportions of spots and sub-spot cell type-specific gene expression profiles to a high accuracy and sensitivity. Importantly, we applied 13 ST datasets in frozen tissues across six cancer types and three ST datasets in formalin-fixed, paraffin-embedded (FFPE) tissues across two

cancer types to define the tumor boundary, deconvolute the cellular composition of the tumor spatial microenvironment (TSME) and reconstruct high-resolution cell type-specific transcriptomics. Further, we combined these data with cell-cell interaction and functional enrichment analysis which revealed that Macro-*SPP1* and Fib-*APSN* are enriched in the tumor boundary across cancers and finally identified potential therapeutic targets in the tumor boundary.

## Results

### Cottrazm: Construct tumor boundary microenvironment based on spatial transcriptomics

Cottrazm aims to construct the microenvironment of tumor boundary based on spatial transcriptomics, single-cell transcriptomics and hematoxylin and eosin (HE)-stained histological images (Fig. 1; see Methods). It consists of three core functions: determining the tumor boundary (*Cottrazm-BoundaryDefine*), deconvoluting spatial transcriptomics (*Cottrazm-SpatialDecon*), and reconstructing a spatial gene expression matrix for sub-spots (*Cottrazm-SpatialRecon*). For *Cottrazm-BoundaryDefine*, we performed the spatial morphological gene expression (SME) normalization algorithm[24], using neighbor information (spatial location) and morphological distance to normalize gene expression of ST data, and obtained a morphologically adjusted expression matrix of spatial transcriptomics. Then, we clustered spots using the K-nearest neighbor (KNN) algorithm based on the morphologically adjusted expression matrix and calculated the normal score based on immune features (see Methods) to obtain values for the cluster within normal tissue (Supplementary Fig. 1a, b). The malignant core was assumed to have the highest CNV scores and composed of the highest proportion of malignant cells. CNV scores were used to separate malignant cells from non-malignant cells in single cell RNA-seq[13–16], Cottrazm then identified clusters with greatest copy number variation by InferCNV[15] as the core spots of malignant spots (Supplementary Fig. 1c, d). Cottrazm arranged spatial spots on hexagonal lattices and calculated the Manhattan distance between spots, and thereby identified the neighboring spots of the tumor core where the Manhattan distance was less than a liner model fitted radius (see Methods). Furthermore, according to the uniform manifold approximation and projection (UMAP) distance to tumor centroid, Cottrazm infers layer by layer from core spots of malignant cells using the hexagonal system to determine the identity of a spot as malignant (Mal) spots or tumor boundary (Bdy) spots. When all neighbors of malignant spots are not malignant, the extrapolation process is exhausted and the remaining spots are labeled as non-malignant regions (nMal), which are neither Mal spots nor Bdy spots.

For *Cottrazm-SpatialDecon*, Cottrazm generated 1) a signature score matrix: specific expressed genes from each cell type in scRNA-seq were utilized as signatures and then signature scores for each spot were calculated; 2) enrichment score matrix: the enrichment score of cell types from scRNA-seq in each spot by parametric analysis of gene set enrichment (PAGE) analysis[28], due to it more sensitive to detect a larger number of significantly altered gene sets and suitable for various sequencing platforms[25,28]; and 3) topic: integration of the KNN clustering result and location information of malignant regions, tumor boundary, and nonmalignant regions. Then, cell types for each topic were determined based on the signature score and enrichment score matrices. Finally, the cell type composition of spots was deconvoluted by dampened weighted least squares (DWLS)[29], which can accurately estimate rare cell types and properly adjust the contribution of each gene[29] and have great performance on cell type deconvolution[23].

For *Cottrazm-SpatialRecon*, we reconstructed cell-type specific gene expression profiles (GEPs) at sub-spot level. We calculated the feature weight of each cell type according to the feature's contribution to each cell type in the scRNA-seq reference, then calculated the feature expression of sub-spots containing a certain cell type according to the cell proportion from deconvolution results and feature weight. In

addition, Cottrazm also provides downstream analysis functions, including visualization of the above three components, differentially expressed genes, functional enrichment of tumor boundary, and the identification of therapeutic targets in the tumor boundary.

## Cottrazm determined three distinct spatial regions from ST data

We performed *Cottrazm-BoundaryDefine* to delineate the malignant (Mal) region, tumor boundary (Bdy) and non-malignant (nMal) region of ST data. We used 12 samples across six cancer types from frozen tissue sections and three samples across two cancer types from FFPE tissue sections according to the hexagonal system (Fig. 2a, Supplementary Fig. 1e, f, and Supplementary Table 1), including colorectal cancer[3] (CRC, $n = 3$), breast cancers (BRCA, $n = 2$), clear cell renal cell carcinoma[30] (ccRCC, $n = 1$), hepatocellular carcinoma[11] (HCC, $n = 4$),

intrahepatic cholangiocarcinoma[11] (ICC, $n = 1$), ovarian cancer (OV, $n = 1$), and squamous cell carcinoma (SCC, $n = 1$)[9]. All samples had clear tumor boundaries to distinguish the malignant region and normal tissue regions after extrapolating less than five layers (Supplementary Fig. 1e). Since 10x Visum in FFPE samples is based on probe sets designed to target specific sequences of RNA, unlike 10× Visum in frozen samples which capture mRNA without bias, we also performed *Cottrazm-BoundaryDefine* in FFPE samples. We observed a clear tumor boundary in three samples with FFPE tissue sections, including BRCA ($n = 1$) and ccRCC ($n = 2$, Supplementary Fig. 1f) samples. We further assess the CNV score among the boundary spots, their first inner and outer circle of malignant and non-malignant spots, respectively. We found CNV scores are significantly higher in malignant cell spots, whereas lower CNV scores were observed for boundary spots and non-

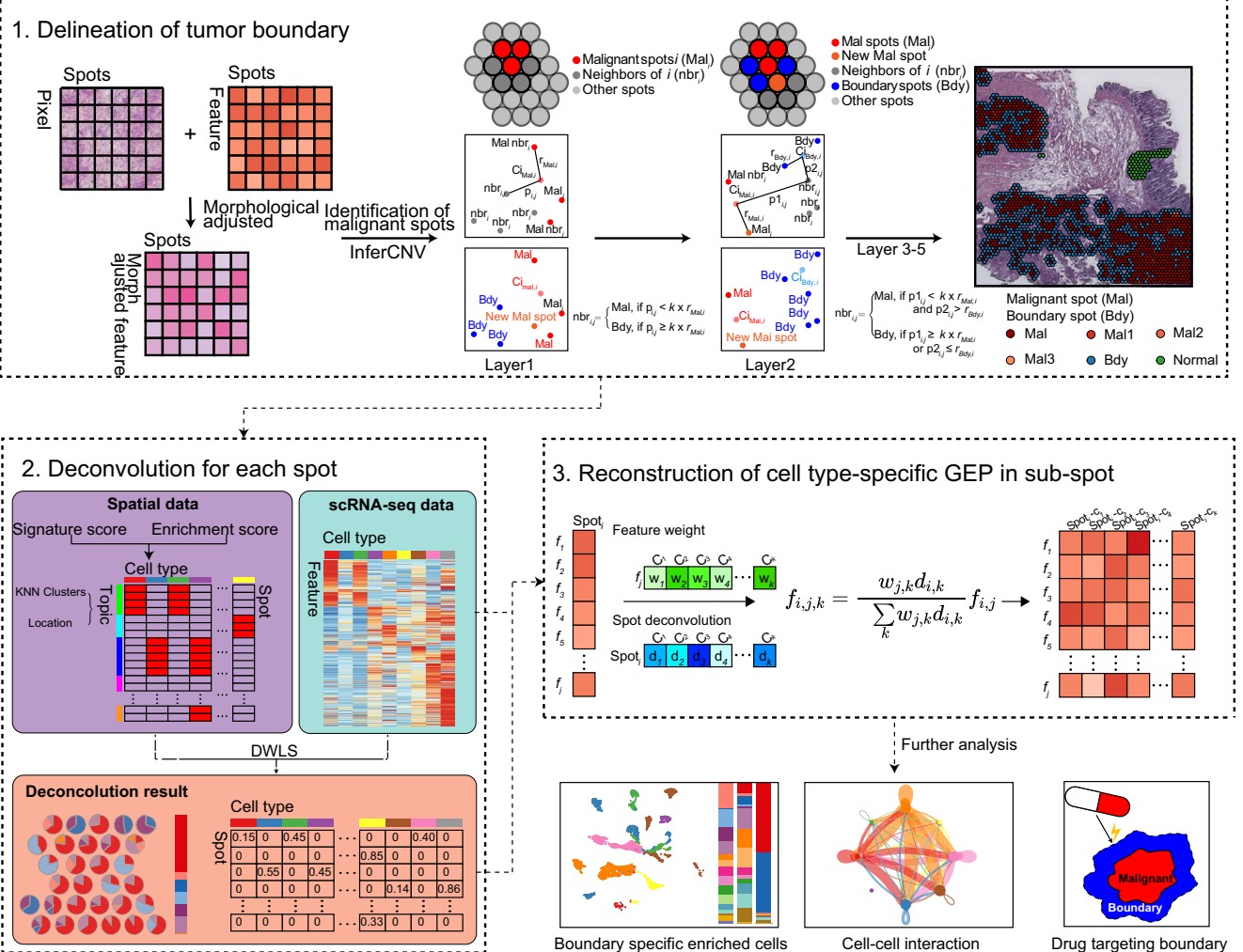

**Fig. 1 | Schematic representation of the Cottrazm workflow.** Step-by-step illustration of the Cottrazm pipeline is as follows: 1. Delineation of tumor boundary. Initially, based on neighboring spot information and morphological distances of HE-staining images, Cottrazm normalizes ST gene expression data to obtain a morphologically adjusted expression matrix. Then clustered by k-nearest neighbor (KNN) algorithm, clusters are identified using copy number variation by InferCNV which define the core spots of malignant cells. Further, Cottrazm arranged spatial spots on hexagonal systems, extrapolated layer by layer from core spots of malignant cells and determined the identity of spots according to UMAP distance to tumor centroid as malignant cell (Mal) spots or tumor boundary (Bdy) spots. When all neighbors of Mal spots are not classed as tumor tissue, the extrapolation is completed. Remaining spots are therefore labeled as nMal. 2. Deconvolution for each spot, Cottrazm generated a signature score matrix from each cell type in the

scRNA-seq dataset. Enrichment score matrices in each spot were then analyzed by parametric analysis of gene set enrichment (PAGE) analysis, and each topic was combined with the KNN clusters and location information. Then, cell types for each topic were determined based on signature score and enrichment score matrices. Finally, the cell type composition of spots was deconvoluted by dampened weighted least squares (DWLS). 3. Reconstruction of cell type-specific gene expression profile (GEPs) at sub-spot level. Features were weighted in each cell type according to the feature contribution in each cell type in the scRNA-seq reference, then feature expression of sub-spots with a certain cell type was calculated by the cell proportion as estimated from the deconvolution results and feature weight. Cottrazm can subsequently be followed by further analysis, including sub-spot GEP analysis, cell-cell interactions or identification of potential druggable targets in the tumor boundary.

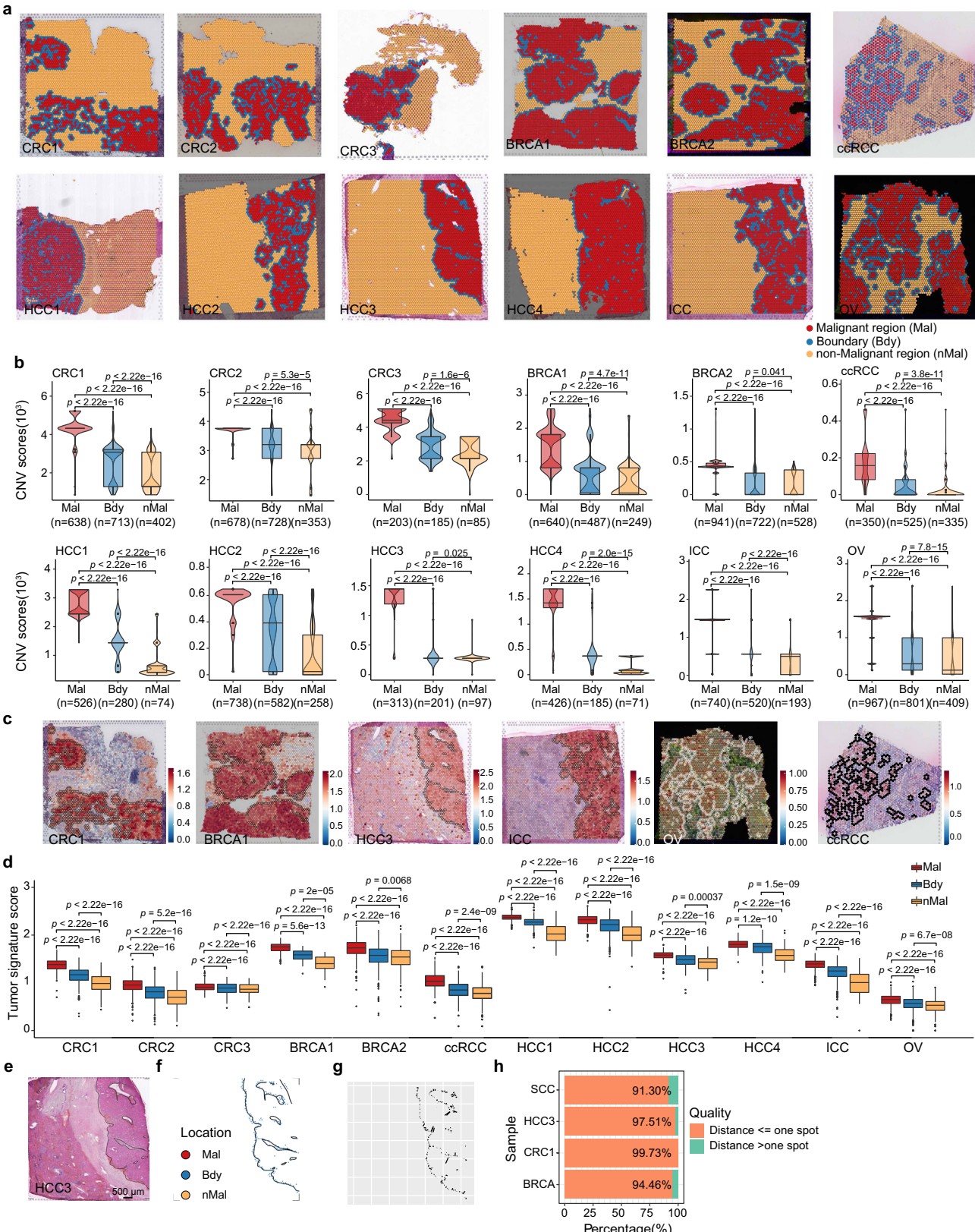

malignant spots (Fig. 2b), suggesting that spots in the tumor boundary contained a low percentage or absence of malignant cells. To validate the characteristics of these three regions, we assessed the tumor signature score of these six cancer types based on the tumor-specific signatures of CRC[3,31–33], HCC[11,34,35], BRCA[36–38], OV[39,40], ICC[35,41,42], and ccRCC[43–45] (see Methods, Supplementary Table 2), and found the

malignant spots had the highest signature scores. Lower scores were observed in the boundary spots, while the non-malignant spots had the lowest signature scores (Fig. 2c, d, Supplementary Fig. 2a), suggesting that our boundary definition is robust.

To validate the concordance of tumor boundary predicted by Cottrazm and annotated by pathologist, We obtained an spatial

**Fig. 2 | The delineation of tumor boundary in multiples cancers. a** Tissue slides were annotated by malignant spots (Mal, red), boundary spots (Bdy, blue), and non-malignant spots (nMal, orange), including colorectal cancer (CRC, $n = 3$), breast cancer (frozen sample, $n = 2$), clear cell renal cell carcinoma (ccRCC, $n = 1$), hepatocellular carcinoma (HCC, $n = 4$), intrahepatic cholangiocarcinoma (ICC, $n = 1$), and ovarian cancer (OV, $n = 1$). **b** Boxplot showing the copy number variations (CNV) score calculated by R package infercnv in three regions defined in **a**. The sample size in each group is labeled in x-axis. **c** Spatial feature plots of signature score of malignant cells in CRC, BRCA, HCC, ICC, OV, and ccRCC. **d** Boxplot showing the tumor signature score in three regions defined in **a**. Sample size for different cancers (CRC1: $n_{Mal} = 963$, $n_{Bdy} = 713$, $n_{nMal} = 2781$; CRC2: $n_{Mal} = 844$, $n_{Bdy} = 728$, $n_{nMal} = 2320$; CRC3: $n_{Mal} = 502$, $n_{Bdy} = 185$, $n_{nMal} = 970$; BRCA1: $n_{Mal} = 2378$, $n_{Bdy} = 487$, $n_{nMal} = 933$; BRCA2: $n_{Mal} = 2208$, $n_{Bdy} = 722$, $n_{nMal} = 1797$; ccRCC: $n_{Mal} = 398$, $n_{Bdy} = 525$, $n_{nMal} = 1084$; HCC1: $n_{Mal} = 969$, $n_{Bdy} = 280$, $n_{nMal} = 1542$; HCC2:

$n_{Mal} = 1125$, $n_{Bdy} = 582$, $n_{nMal} = 2965$; HCC3: $n_{Mal} = 1229$, $n_{Bdy} = 201$, $n_{nMal} = 3328$; HCC4: $n_{Mal} = 2036$, $n_{Bdy} = 185$, $n_{nMal} = 1892$; ICC: $n_{Mal} = 1468$, $n_{Bdy} = 520$, $n_{nMal} = 2666$; OV: $n_{Mal} = 1822$, $n_{Bdy} = 801$, $n_{nMal} = 870$). **e** HE stained images with pathologist-annotated tumor boundary of HCC. Scale bar, 500 μm. **f** Tumor boundary annotated by pathologist (black) and boundary spots annotated by Cottrazm (Blue). **g** Line segments of the shortest distance from boundary spots annotated by Cottrazm to the pathologist's boundary. **h** Bar plot showing the proportion of boundary spots which distance to pathologist's boundary is less than or equal to one spot. A two-sided Wilcoxon signed-rank test was used to assess statistical significance in **b** and **d**. The boxes in **b** and **d** show the median ±1 quartile, with the whiskers extending from the hinge to the smallest or largest value within 1.5× the IQR from the box boundaries. Source data are provided as a Source data Fig. 2a–d.

---

transcriptomics of SCC performed by 10X Genomics platform and the HE staining with leading edge annotated by pathologist in previous study[9] (Supplementary Fig. 2b). Then we asked the professional pathologist to help us annotate the tumor boundary in representative HE staining which used in our original study, including BRCA, CRC, and HCC (Fig. 2e, Supplementary Fig. 2b). Then, we imported the outlined tumor boundary layer into R and converted into an unordered set of points (pixel coordinates), each point defining a position on the tumor boundary layer. Further, we calculated pairwise distances between spots of ST and tumor boundary outline by KNN (dbscan R package) to find the minimum distance from a spot to the closest outline (Fig. 2e–g, Supplementary Fig. 2b–d). We considered the tumor boundary spot predicted by our tool Cottrazm is match to the location of pathologist-annotated tumor boundary outline when their distance is less than one spot size. Our result showed highly consistency between predicted tumor boundary spot and pathologist-annotated tumor boundary outline, ranging from 91.30% in SCC to 99.73% in CRC (Fig. 2h), suggesting the great performance of our tool on the delineation of tumor boundary.

## Cottrazm deconvolutes cellular composition and reconstructs cell type-specific gene expression profiles at sub-spot level

To evaluate the performance of *Cottrazm-SpatialDecon*, we created a simulated mixture of cells with known cell-type labels from a CRC scRNA-seq dataset, based on the spatial locations obtained from *Cottrazm-BoundaryDefine* and characteristics of cell composition in the TME (see Methods). We benchmarked *Cottrazm-SpatialDecon* against other published ST deconvolution tools, including Stereoscope[46], Cell2Location[47], RCTD[48], SpatialDWLS[49], STRIDE[50], and SPOTlight[51], as well as methods for bulk RNA-seq, including CIBERSORTx[52] and MuSiC[53]. We assessed the performance on simulated ST data by these deconvolution tools: the accuracy of Cottrazm was > 0.75 across T cells, fibroblast cells, B/Plasma cells, endothelial cells, normal epithelial cells and myeloid cells, respectively, reaching 0.98 in malignant cells (Fig. 3a). The F1 score of *Cottrazm-SpatialDecon* ranged from 0.67 in normal epithelial cells to 0.98 in malignant cells, and specificity ranged from 0.62 in T cells to 1 in fibroblast cells (Fig. 3a). These performance parameters suggested *Cottrazm-SpatialDecon* can correctly classify cell types with high recall/sensitivity and precision, and can correctly predict the absence of cell types in spots. Cell type predictions of SpatialDWLS and RCTD show comparable accuracy, F1 score, and specificity (Fig. 3a). We performed Spearman's correlation on cell type-level and spot-level data comparing the prediction of *Cottrazm-SpatialDecon* and actual labels in the simulated dataset. The Spearman correlation coefficient ($R$) value of *Cottrazm-SpatialDecon* prediction and true proportion ranged from 0.55 in T cells to 0.92 in malignant cells (Fig. 3b), indicative of accurately distinguishing cell types. The median Spearman's correlation coefficient ($R$) of *Cottrazm-SpatialDecon* prediction and true proportion of spots is 0.88, indicating the predicted value of *Cottrazm-*

*SpatialDecon* had the highest concordance with the ground truth in cell composition of spots, and outperformed Stereoscope (median $R = 0.81$), and Cell2Location (median $R = 0.83$), SpatialDWLS (median $R = 0.86$), CIBERSORTx (median $R = 0.81$) and RCTD (median $R = 0.80$; Fig. 3c). In addition, we used Pearson correlation coefficient value to assess the concordance of prediction by Cottrazm and other eight deconvolution tools and true proportions, we obtained *Cottrazm-SpatialDecon* have high performance, and Stereoscope, Cell2Location, RCTD[48], SpatialDWLS, and STRID these tools have comparable performance to Cottrazm in correlation analysis (Supplementary Fig. 3a, b). Further, we assessed the similarity between the prediction and the ground truth of cell types in each spot by Jensen–Shannon Divergence (JSD) distance metric, where a smaller value indicates a higher similarity between the predicted and true dataset (Fig. 3d). *Cottrazm-SpatialDecon* achieved median JSD values of 0.089 which are comparable with Cell2Location, SpatialDWLS, RCTD, and CIBERSORTx of 0.032, 0.07, 0.061, and 0.067, respectively. These consistently superior performance metrics demonstrated the enhanced accuracy and robustness of *Cottrazm-SpatialDecon*.

To evaluate the performance of *Cottrazm-SpatialRecon*, we obtained an additional independent CRC scRNA-seq cohort[54] as training dataset for an independent validation and executed *Cottrazm-SpatialRecon* to obtain GEPs at sub-spot level of CRC simulated mixtures that we created above. Both the prediction of *Cottrazm-SpatialRecon* and the true cellular composition of simulated mixtures contain seven distinct cell types, including T cells, fibroblast cells, B/Plasma cells, endothelial cells, normal epithelial cells, myeloid cells, and malignant cells (Supplementary Fig. 3c). We integrated the sub-spot matrix for prediction and truth and found consistent cell type clustering results (Fig. 3e, f). We performed Spearman's and Pearson's correlation based on the predicted results and the true average expression of each gene in each sub-spot cell type: Spearman's correlation coefficient ($R_S$) and Pearson's correlation coefficient ($R_P$) of *Cottrazm-SpatialRecon* prediction and true proportion ranged from 0.77 in endothelial cells to 0.86 in malignant cells and myeloid cells, 0.81 in T cells to 0.98 in myeloid cells, respectively (Fig. 3g, h). The results suggest that the prediction of *Cottrazm-SpatialRecon* can reflect cell type-specific gene expression profiles at the sub-spot level with high fidelity and high-resolution.

## Cottrazm characterizes cellular composition along the malignant-boundary-non-malignant spatial axis

To demonstrate the efficient application of Cottrazm to real spatial transcriptomics data and dissect the spatial microenvironment along the malignant-boundary-non-malignant axis, we have classified CRC tissues into malignant, tumor boundary, and non-malignant regions through *Cottrazm-BoundaryDefine*. We then deconvoluted the cellular composition of CRC tissues in these three regions by *Cottrazm-SpatialDecon* and compared different compositions among the regions (Fig. 4a). We found that malignant spots typically consisted of

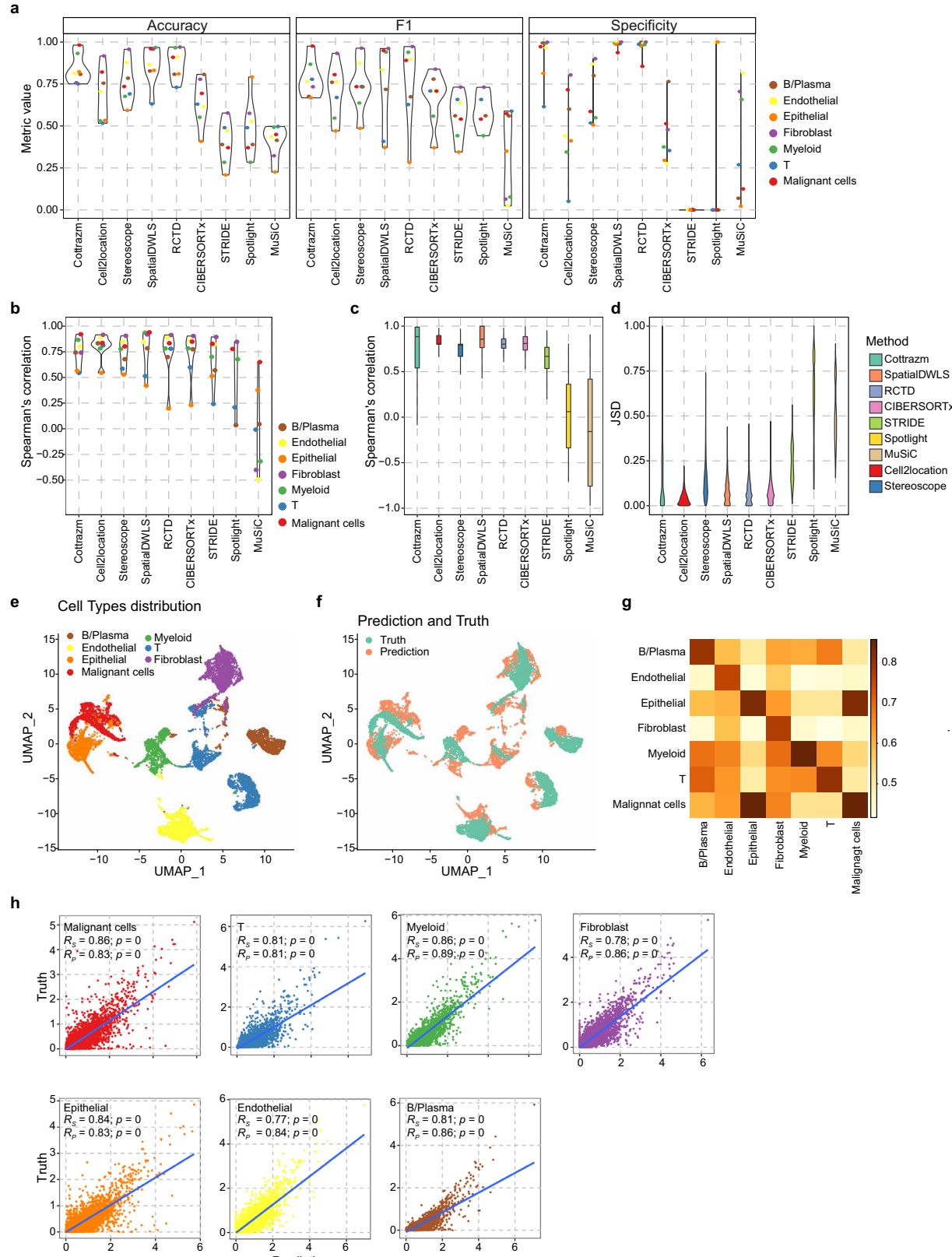

malignant cells, but were absent of T cells or other cell infiltration. The boundary spots consisted of myeloid cells, fibroblast cells, and endothelial cells, whereas non-malignant spots excluded malignant cells and had various cellular composition characteristics depending on the tissue location (Fig. 4a, b). Tertiary lymphoid structures (TLSs) are ectopic lymphoid organs that develop in non-lymphoid tissues at sites of chronic inflammation, including tumors, and represent a promising avenue for cancer immuotherapy[30,55–59]. Spots in TLSs consisted of a greater percentage of T cells and B/Plasma cells, and lower percentage of stromal cells and myeloid cells (Fig. 4a, b). The normal epithelium tissue-related spots had a high proportion of epithelial cells and fewer immune cells and fibroblast cells, while stromal-related spots had high

**Fig. 3 | Benchmarking Cottrazm's performance of deconvolution and reconstructions using simulated data. a** Benchmarking classification performance of Cottrazm and other eight deconvolution tools on simulated mixtures, including accuracy, F1 score, and specificity. **b** A benchmark of the ability to distinguish different cell types across different deconvolution tools. Spearman's correlation was performed to evaluate the correlation between the predicted proportions and the ground truth for each cell type. **c** Benchmark of deconvolution tools' consistency of cell type distribution between the predicted proportions and the ground truth for each spot. The box plot reflects the overall distribution of Spearman's correlation calculated in each simulated spot ($n = 2700$) for each method. The boxes show the median ±1 quartile, with the whiskers extending from the hinge to the smallest or largest value within 1.5× the IQR from the box boundaries. **d** Proportion prediction performance of the different deconvolution

tools on simulated mixtures by Jensen−Shannon Divergence (JSD). **e**, **f** UMAP projections of cell type specific gene expression profiles (GEP) at sub-spot level by integrating the predicted proportions and the ground truth of simulated mixtures. **e** colored by cell types, **f** colored by the prediction (orange) and the truth (green). **g** Heatmap showing the concordance between cell type proportions measured by Cottrazm and the ground truth of simulated mixtures by Spearman's correlation. **h** Scatter plots depicting concordance between cell type proportions measured by the two-sided Spearman's ($Rs$) and Pearson's ($Rp$) correlation coefficient of Cottrazm and the ground truth of simulated mixtures, including tumor cells, T cells, myeloid cells, fibroblast cells, endothelial cells, and B/Plasma cells. UMAP Uniform Manifold Approximation and Projection. Source data are provided as a Source data Fig. 3a–h.

percentages of fibroblast cells and endothelial cells, and fewer immune cells (Fig. 4a, b). We further deconvoluted the cellular composition in other cancer types using frozen or FFPE tissues, including HCC, BRCA, and OV (Supplementary Fig. 4a–d). There was a high fibroblast, macrophage, and endothelial cell component at the tumor boundary, suggestive of cellular composition commonalities in tumor boundaries across cancer types. To further validate the location of T cells, we compared CD3 staining of invasive ductal carcinoma (IDC) tissue section of breast cancer (Supplementary Fig. 4e), we obtained the processed image which highlight CD3 (cyan) staining from a previous study[26]. Then, we compared deconvolution results of Cottrazm predicted T with CD3 staining in IDC sample. We found the location of predicted T cells by Cottrazm was highly correlated with CD3 fluorescence ($R = 0.56$, $p < 2.2e−16$, Supplementary Fig. 4f–i). In addition, we examined the expression of CD3-encoding genes of the IDC tissue and found consistency with the CD3 fluorescence (Supplementary Fig. 4k). This suggests the tumor boundary acts as a barrier, lymphocytes such as T and B cells are restricted from infiltrating into the malignant region, thus a microenvironment of immune exclusion is formed. Nevertheless, further characterization of cellular composition in various cancer types with larger sample size is necessary.

To refine the subpopulation characteristics of the above cell types, we defined sub-spot as different cell types in each spot, and further performed *Cottrazm-SpatialRecon* to reconstruct cell type-specific GEPs at sub-spot level using ST data from multiple cancer types. We integrated sub-spot GEPs data for different tumor samples with the harmony algorithm[60]. In total, we obtained 15,555 sub-spots in three CRC ST samples, and 43,813 sub-spots in four HCC ST samples. We performed graph-based clustering based on harmony-corrected principal components and annotated each cluster with their bio-markers and visualized by the UMAP analysis. We found that the different cell types of the deconvolution can be significantly separated by sub-spot GEPs and that all cell types were present in all tumor samples (Fig. 4c, Supplementary Fig. 5a), suggesting concordance of *Cottrazm-SpatialDecon* and *Cottrazm-SpatialRecon*. We further annotated the cell subtypes based on sub-spot GEPs. Myeloid cells can be divided in to three subtypes in CRC tissues, including monocyte/dentritic cells (Mono/DC) defined by *CLEC10A*, *S100A8*, and *S100A9* expression, *FOLR2*⁺ macrophages (Macro-*FOLR2*), and *SPP1*⁺ macrophages (Macro-*SPP1*) (Fig. 4d, Supplementary Fig. 5b, c). Compared with malignant and non-malignant regions, Macro-*SPP1* was significantly enriched in the tumor boundary, while Macro-*FOLR2* tended to be enriched in non-malignant region and Mono/DC cells tended to be enriched in the region surrounding the TLS (Fig. 4e–g, Supplementary Fig. 5d, e). To further explore the biological function of differential localization of different macrophages, we found that Macro-*SPP1*, which localized along the boundary, mainly functionally enriched in cell migration and metabolic, while Macro-*FOLR2*, localized around TLS in the non-malignant region, mainly functionally enriched in antigen presentation (Supplementary Fig. 5f). The functional phenotypes of tumor associated macrophage in pan-cancer can be defined as dichotomous

functional phenotypes, including angiogenesis and phagocytosis[61,62]. Using the angiogenic and phagocytic signatures, we assessed the functional phenotypes of two macrophage subtypes with differential localization. We found angiogenesis signature genes preferentially expressed in Macro-*SPP1*, while phagocytosis signature genes enriched in Macro-*FOLR2* (Supplementary Fig. 5g). In addition, Macro-*SPP1* showed significantly higher M2 score than Macro-*FOLR2* consistent with previous study[63], suggesting the key roles of Macro-*SPP1* in the tumorigenesis (Supplementary Fig. 5h).

To validate the differential distribution of myeloid cell subtypes in other cancer types, we classified subtypes of myeloid cells according to the clustering of sub-spot GEPs in HCC tissues, including monocytes, *TGFB1*⁺ macrophages (Macro-*TGFB1*), Macro-*SPP1*, and *MARCO*⁺ macrophages (Macro-*MARCO*) (Fig. 4h, Supplementary Fig. 5i–l). Macro-*SPP1* was also significantly enriched in the boundary region in HCC, while other myeloid subtype patterns were distinct from CRC. The non-malignant regions of HCC were tended to be enriched of Macro-*MARCO* macrophages while the non-malignant regions of CRC were Macro-*FOLR2* enriched (Fig. 4e–g and i, k, Supplementary Fig. 5k–l). Macro-*TGFB1* tended to be enriched in the malignant region of HCC, but this macrophage subtype was not identified in CRC (Fig. 4i, k). We speculated that Macro-*SPP1* is a critical cellular composition in the tumor boundary, but other subtypes of macrophages were not consistently enriched in malignant or non-malignant regions. To validate this hypothesis, we further examined the signature score of Macro-*SPP1* in BRCA, ccRCC, and OV, and found consistent results suggesting that Macro-*SPP1* was significantly enriched in boundary regions in the above cancer types (Supplementary Fig. 5m).

To explore the differences in fibroblast subtypes along the malignant-boundary-non-malignant axis, we stratified fibroblasts into *APSN*⁺ fibroblasts (Fib-*APSN*), *SFRP2*⁺ fibroblasts (Fib-*SFRP2*), and myofibroblast (Myofib) which express the *MYLK* marker (Fig. 4l, Supplementary Fig. 5c). We identified significant enrichment of Fib-*APSN* and Fib-*SFRP2* in tumor boundary and non-malignant regions, respectively (Fig. 4m–o, Supplementary Fig. 6a, b). Myofib enrichment tended towards non-malignant spots (Fig. 4m–o). For the function roles of fibroblast subtypes Fib-*APSN* and Fib-*SFRP2*, we found that Fib-*ASPN*, which tends to localize along the tumor boundary, is highly active in pathways that contribute to the formation of desmoplastic structures, including extracellular matrix assembly, collagen fibril organization, and collagen biosynthetic process (Supplementary Fig. 6c), while Fib-*SFRP2* localized in the non-malignant region functionally enriched in collagen degradation and mesenchyme migration. To validate the fibroblast subtype Fib-*APSN*, we explored fibroblast subtypes in HCC and found Fib-*APSN* was also tended to enrich in boundary region of the HCC samples, therefore fibroblast phenotype within the non-malignant region was disease-specific (Fig. 4p–s, Supplementary Fig. 6d, e). Furthermore, the Fib-*APSN* signature score was enriched in the boundary region of BRCA, OV, and ccRCC samples (Supplementary Fig. 6f). In addition, we obtained samples with

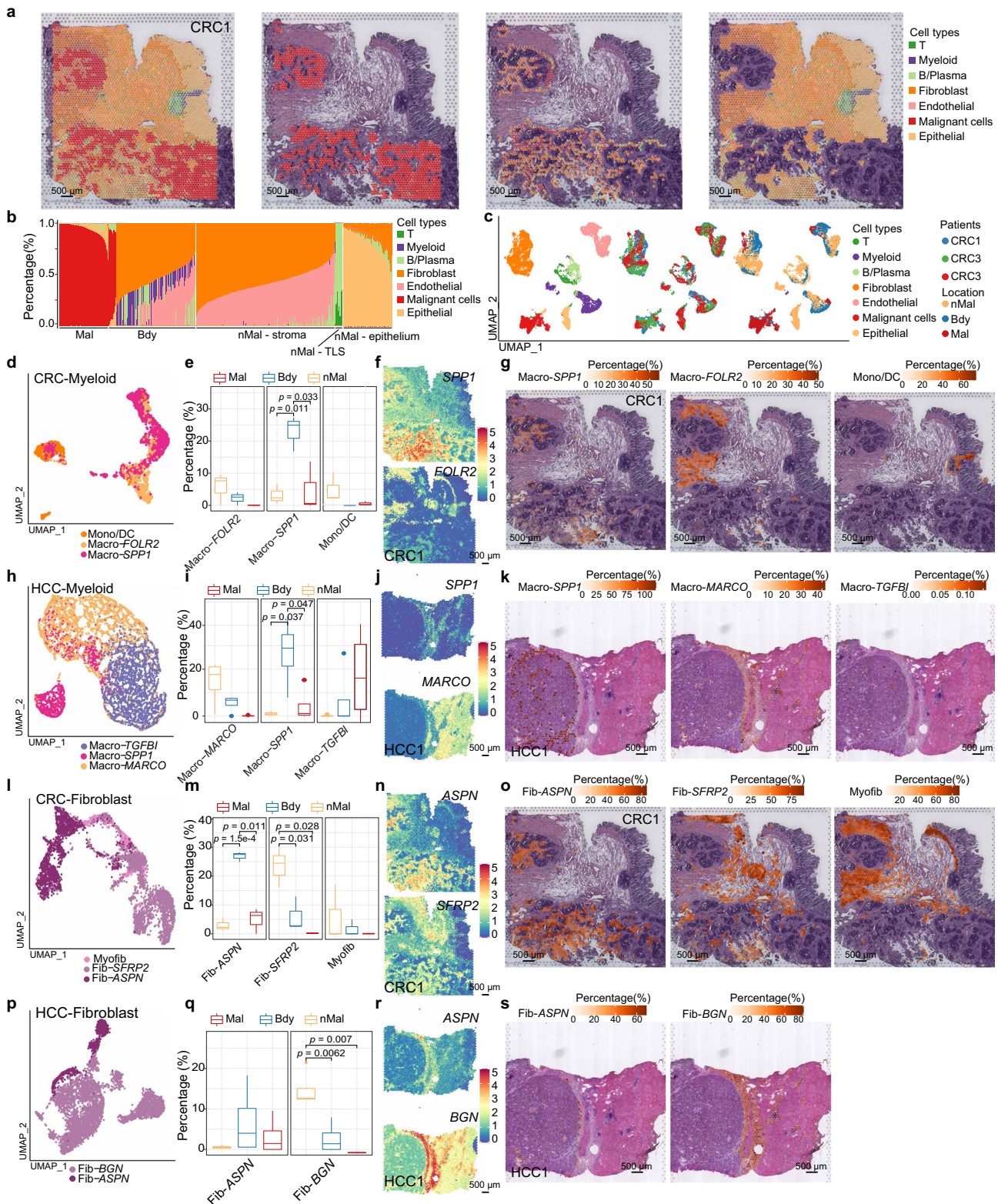

immunostaining from human protein atlas (HPA)[64], and found the stained sample from CRC patient (id: 3264) we found the ASPN stain tended to enrich at tumor boundary, while POSTN, representative marker of another subtype of fibroblasts, tended to enrich at tumor stroma (Supplementary Fig. 6g). Consistent results were observed in the sample of BRCA patients (id: 1939, Supplementary Fig. 6h). These results demonstrate that Cottrazm can spatially dissect the cellular composition and cell-type signatures along the malignant-boundary-non-malignant axis. We revealed that Macro-*SPP1* and Fib-*APSN* are preferentially enriched in the tumor boundary across cancer types.

## Crosstalk of Macro-*SPP1*, Fib-*ASPN*, and malignant cells in the tumor boundary contribute to immune exclusion

To investigate differential biological processes occurring at the tumor boundary niche, we performed enrichment analysis using boundary-specific gene expression to identify extracellular matrix (ECM)

**Fig. 4 | Spatial distribution of cell types along the malignant-boundary-nonmalignant axis. a** Spatial scatter pie plots representing the proportions of the seven cell types predicted by Cottrazm in whole CRC ST slide, malignant spots, boundary spots, and nonmalignant spots. Scale bar, 500 μm. **b** Bar plots representing the proportions of the seven cell types predicted by Cottrazm in each spot. **c** UMAP projections of sub-spots in three CRC ST dataset predicted by Cottrazm, each dot denotes one sub-spot; color represents cluster origin (left panel), patient donors (middle panel), and the region of origin (right panel). **d**–**s** The characteristics of myeloid subtypes and fibroblast subtypes in the ST dataset. The UMAP projections of subtypes of myeloid cells in CRC (**d**) and HCC (**h**), subtypes of fibroblasts in CRC (**i**) and HCC (**p**). Box plots showing proportion of myeloid cell subtypes of CRC (n = 3, **e**) and HCC (n = 4, **i**), fibroblast subtypes of CRC (n = 3, **m**) and HCC (n = 4, **q**) in each region. Spatial feature plots showing the expression of *SPP1* and *FOLR2* in CRC (**f**), *SPP1* and *MARCO* in HCC (**j**), *ASPN* and *SFRP2* in CRC (**n**), *ASPN* and *BGN* in HCC (**r**). Predicted proportion within each capture spot for Macro-*SPP1*, Macro-*FOLR2*, and Mono/DC in CRC (**g**), Macro-*SPP1*, Macro-*MARCO*, and Mcro-*TFGB1* in HCC (**k**), Fib-*ASPN*, Fib-*SFRP2*, and Myofib in CRC, Fib-*ASPN* and Fib-*BGN* in HCC (**s**). Color indicates the percentage of cell type. Scale bar, 500 μm in **g**, **k**, **o**, and **s**. The boxes in **e**, **i**, **m**, and **q** show the median ±1 quartile, with the whiskers extending from the hinge to the smallest or largest value within 1.5× the IQR from the box boundaries. A two-sided unpaired t-test was used to assess statistical significance in **e**, **i**, **m**, and **q**. Source data are provided as a Source data Fig. 4a–d, h, i, l–m, p–q.

organization, collagen fibril organization, cell-substrate adhesion, positive regulation of chemotaxis, and response to TGF-β, which contributes to the formation of desmoplastic structure[3,65–69]. These were enriched in both CRC and HCC tumor boundaries (Fig. 5a, b, Supplementary Fig. 7a, b), suggesting that ECM related pathways may promote tumor boundary structure formation. We further identified the top 100 specifically expressed genes in the tumor boundary of the three CRC ST samples which overlapped significantly (Supplementary Fig. 7c), and included *SPP1*, *MMP11*, and *COL1A1* (Fig. 5a). Pathway enrichment for these top 100 genes also highlighted ECM organization, further implicating the ECM in tumor boundary formation (Supplementary Fig. 7d). To fully dissect enrichment of genes and pathways in the tumor boundary, we evaluated cell-cell interactions in the tumor boundary niche, including the boundary spots, their first inner circle of malignant regions and first outer circle of non-malignant region, therefore encompassing the tumor boundary and immediate surroundings (Fig. 5c, d). Integrated analysis by *Cottrazm-BoundaryDefine* defined the tumor boundary and *Cottrazm-SpatialRecon* reconstructed subspot GEPs. We identified Macro-*SPP1* and Fib-*APSN*, the major cellular constituents of the tumor boundary, which exhibited strong interactions with each other and significantly with tumor cells (Fig. 5e, f). In addition, *PLVAP*⁺ endothelial cells (Endo-*PLVAP*) were enriched in the tumor boundary and exhibited interactions with Macro-*SPP1*, Fib-*APSN*, and tumor cells. By contrast, other cell types beyond the tumor boundary, such as B cells, Macro-*FOLR2*, Myofib, and Fib-*SFRP2*, exhibited weak interactions in the tumor boundary niche (Fig. 5e, f). We further explored the top regulatory signaling pathways in the malignant cells, as well as in Macro-*SPP1* and Fib-*APSN*, and established Macro-*SPP1* as the main sender of the SPP1 signaling pathway. Fib-*APSN* was identified as the main secretor of collagen, and malignant cells were the primary senders of the MIF signaling pathway (Fig. 5f, g). Cottrazm identified various ligand-receptor signaling pathways and ECM components as key interaction partners between the various tumor boundary constituents (Fig. 5g).

The ST samples that were investigated typically showed low levels of lymphocyte infiltration in malignant regions (Fig. 5h, Supplementary Fig. 7e). Since the Macro-*SPP1* and Fib-*APSN* enriched tumor boundaries contribute to the formation of desmoplastic boundary structure related to tumor immune excluded microenvironment (TIEM), these cells potentially limit T cell infiltration into malignant regions. To test this hypothesis, we examined the distribution of T cell infiltration and found that killer T cells (T-*KLRB1*) were significantly enriched outside malignant regionswhile exhausted T cells (T-*HAVCR2*) were enriched at the tumor boundary (Fig. 5i); the infiltration of these cells is limited to the malignant region. To determine the characteristics of TIEM, we assessed the level of immune exclusion in tumor through using the gene signature score on the TIEM-specific tumor boundary called the immune excluded score (ieScore). We calculated ieScore by gene set variation analysis (GSVA)[70] (see Methods) in each patient of the CRC cohort from The Cancer Genome Atlas (TCGA)[71], and stratified patients into ieScore high and low groups. High ieScore patients correlated with lower infiltration of lymphocytes and CD8⁺T cells (Fig. 5j). Furthermore, we

performed survival analysis and observed that high ieScore was associated with worse overall survival (log-rank test, p = 0.039) and progression-free survival (log-rank test, p = 0.04, Fig. 5k). These results suggest that the specific tumor boundary signature in cancer patients associated with worse prognosis and tumor immune exclusion. Based on the identification of 37 up-regulated and 20 down-regulated genes in the tumor boundary compared to other regions, we queried this signature and subsequently mapped the genes to the Connectivity Map (CMAP)[72]. Using this method, we identified candidate genes which are known to be druggable, for example genes with the highest negative connectivity score from the mapped drugs included TGF beta receptor which is coupled to the inhibitor D-4476 (Supplementary Fig. 7f). These drugs could be potentially used to target the tumor boundary to disrupt the barrier structure and enhance T cell infiltration into malignant regions, thereby increasing patient susceptibility to immunotherapy.

## Discussion

Tumor tissues have complex spatial architecture, and the rigorous analysis of the spatial microenvironment is critical to understanding tumorigenesis, progression mechanisms and discovery of therapeutic targets. However, most ST tools are developed based on ST datasets from developmental tissues (e.g. mouse brain posterior brain, human heart)[27,49,51,73,74], which have relatively clear annotation of cell types. ST tools developed specifically for application within the tumor microenvironment are lacking. In this study, we present Cottrazm, a tool specific for tumor spatial microenvironment analysis, to determine the tumor boundary, deconvolute cellular composition, and reconstruct cell type-specific gene expression profiles at sub-spot level by integrating spatial transcriptomics with scRNA-seq data and histology imaging. Furthermore, we utilized our method to demonstrate the biologic processes and cell-cell interactions at the tumor boundary thus identifying multiple potential therapeutic targets in this local microenvironment.

We have presented *Cottrazm-BoundaryDefine* for the definition of tumor boundary. Through evaluation of CNV and tumor expression characteristics, cross-validation with the cellular composition deconvoluted by *Cottrazm-SpatialDecon*, tumor boundary spots can be identified with high accuracy. SNV burden is more pronounced than the CNV burden in some cancer types (e.g., melanoma), CNV can still distinguish malignant regions from Bdy and nMal regions by Cottrazm (Supplementary Fig. 1e). In future work, the more malignant cells-related factors may need be involved to assess the tumor boundary. In the *Cottrazm-SpatialDecon* section, Cottrazm applied DWLS[29] to estimate the cell-type composition from gene expression, compared to SpatialDWLS[49], we combined the signature score matrix and enrichment score matrix to determine signatures of cell types for each topic. In addition, we integrated the location information into topics to distinguish the different characteristics among malignant regions, tumor boundary, and non-malignant regions, which will improve the accuracy of cell type selection. *Cottrazm-SpatialRecon* enables robust gene expression profiling of cell types at sub-spot level. In our study, we reconstruct high-resolution profiling of gene expression in sub-spots

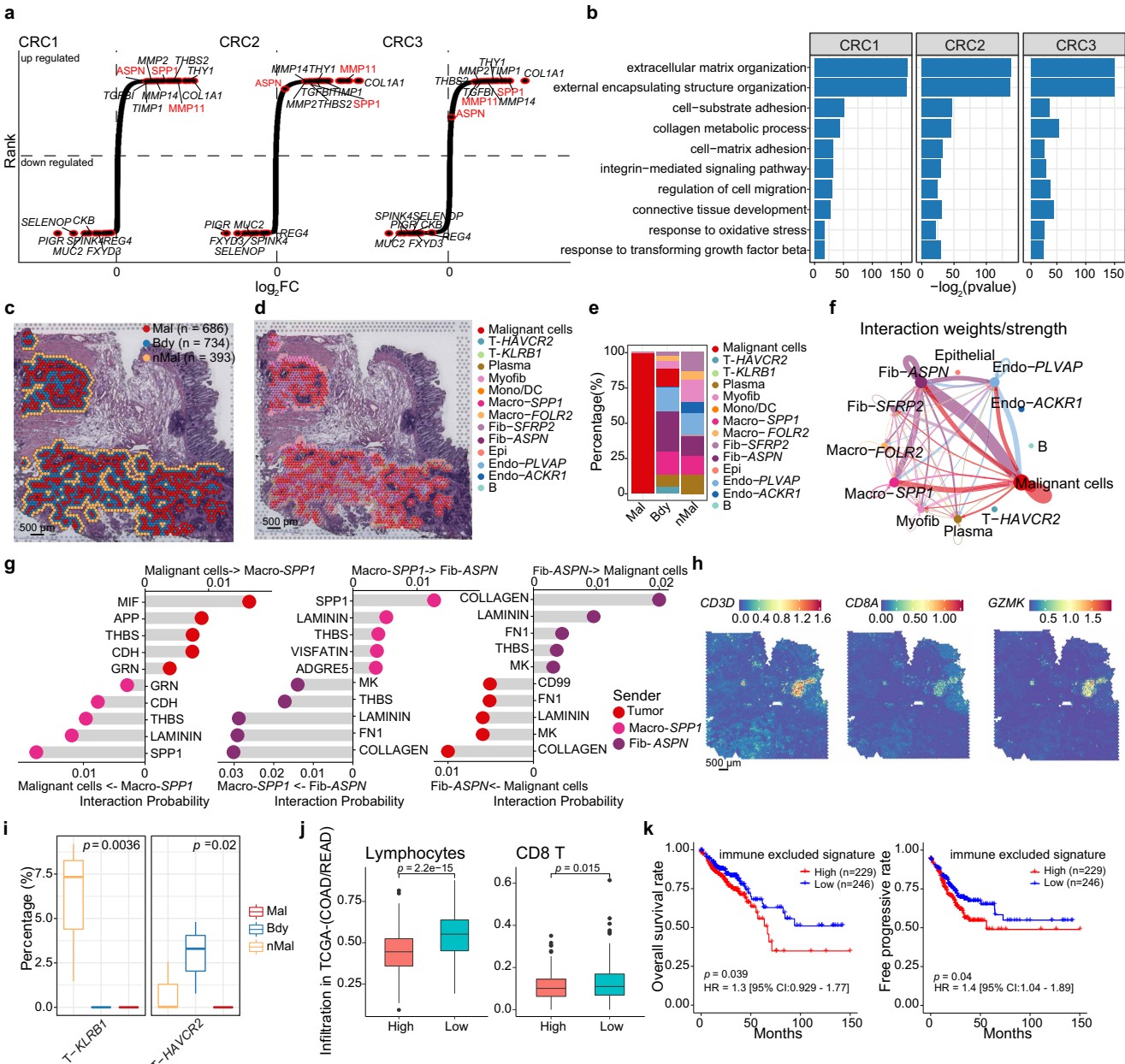

**Fig. 5 | Characterization of microenvironment in tumor boundary. a** Rank-ordered plot showing log2 fold change between gene expression in boundary spots and other spots in three CRC ST samples. **b** Gene ontology (GO) terms of genes significantly enriched in boundary spots of three CRC ST samples. The statistical analysis was performed by two-sided Fisher's test. The location (**c**) and cellular composition in Pie plot (**d**) for boundary spots (blue), and its first inner circle for malignant spots (red) and first outer circle for non-malignant spots (orange). **e** Bar plots showing proportion of cell types in each region defined in **c**. **f** Circle plot visualizing the cellular communication of cell types in three regions defined in **c**. Circle sizes are proportional to the number of cells in each cell group and thickness of the flow represents the interaction weight. **g** Lollipop plot showing the interaction probability for signaling pathway between sender and receptor cell types. **h** Spatial feature plot showing the expression of *CD3D*, *CD8A*, and *GZMK* in CRC1 sample. **i** Box plot showing the proportion of T cell subtypes in malignant spots (Mal, *n* = 3), boundary spots (Bdy, *n* = 3), and non-malignant spots (nMal, *n* = 3). **j** Box plot showing the infiltration of lymphocyte and CD8+ T between immune excluded score (ieScore) high (red, *n* = 224) and low (green, *n* = 228) group, which stratified by median ieScore. The boxes in **i** and **j** showed the median ±1 quartile, with the whiskers extending from the hinge to the smallest or largest value within 1.5× the IQR from the box boundaries. **k** Kaplan−Meier curves indicated the correlation of ieScore and overall survival time (left) and progression-free survival time (right) in the TCGA CRC cohort. A two-sided Wilcoxon signed-rank test was used to assess statistical significance in **a**, **i** and **j**. A two-sided log- rank test was used in **k**. Scale bar, 500 μm in **c d**, and **h**. Source data are provided as a Source data Fig. 5a–g, i.

levels of tumor ST data to help distinguish cell-cell interaction in the same spot. This function is introduced in Cottrazm to help in-depth dissection of the tumor spatial microenvironment. These results demonstrate the advantages of Cottrazm in dissecting the tumor spatial microenvironment, regardless of whether samples were processed as frozen or FFPE tissues as in the 10X Genomics Visium-based ST datasets. However, Cottrazm predicted tumor boundary is composed of spatial spots, the thickness of predicted boundary is correlated with the size of spot size of different technology. In Cottrazm, the diameter of spatial spots and the distance of neighbored spots were relatively determined by the spot size on sequencing slide and the resolution of image. Relative diameter can be applied to variable sequencing resolution while the absolute distance needs to be re-adjusted according to different resolution of image and spatial spot,

which has certain limitations. With the advantage of ST technology, most ST datasets have high resolution and spot of tumor boundary contain few cells. We will continue to focus on tumor ST data generated via other technologies with higher resolution to further improve the performance of our tool on multiple ST platforms.

Despite the presence of tumor-infiltrating lymphocytes in HCC, CRC, OV, and BRCA samples, these cancer types have very low response rates to ICB therapy[75–80]. Most ST datasets show low infiltration of lymphocytes in the malignant region and the T cells around tumor boundary express exhaustion markers (Fig. 5i and Supplementary Fig. 4). Cottrazm combines functional enrichment and cell-cell interaction analysis, and identified that Macro-SPP1 and Fib-APSN cells were associated with worse prognosis. These cell types accumulate proximal to, and interact with, malignant cells, inducing ECM secretion which promotes the barrier structure to maintain separation of malignant and immune cells, thus protecting malignant cells from CD8+ killer T cells. The enrichment of Macro-SPP1 or Fib-APSN cells at the tumor boundary may be a common characteristic in ICB-insensitive cancers, that deserve further investigations.

In summary, Cottrazm provides a computational framework for ST data in the TSME. We have described the demonstration of Cottrazm to analyze spatial location, cellular composition, gene expression profiling, and mining of biological process, cell interactions, and potential therapeutic targets in the tumor boundary. Our package offers an approach for TSME analysis based on the malignant-boundary-non-malignant axis. From a translational perspective, Cottrazm can accurately estimate the cellular composition and gene expression profiles in sub-spots to provide potential therapeutic strategies through targeting of regulatory factors in specific local microenvironments.

## Methods

### Implementation of Cottrazm

#### Cottrazm-BoundaryDefine: Delineation of tumor boundary

**Morphologically adjusted expression matrix of spatial transcriptomics.** Cottrazm takes expression matrix of spatial transcriptomics (ST) and HE-stained histological image as input. Spatial gene expression data is stored in an M × N matrix with M spots and unique molecular identifier (UMI) counts for N genes, and the (x,y) two-dimensional (2D) spatial coordinates of each spot. Based on spots 2D spatial coordinates, the HE-stained histological images were extracted and converted to an M × D matrix with M spots and 2048 pixel utilizing ResNet50, a well-established convolutional neural network (CNN) model for image classification in computer vision and ImageNet, a dataset containing millions of images. We performed principal component analysis (PCA) to reduce spatial gene expression and pixel matrix are to 50 PCs, respectively. We employ SME normalization algorithm from *Stlearn*[24] to adjust gene expression according to the spot image matrix and obtain morphologically adjusted gene expression matrix (Morph).

**Identification of core spots of malignant cell.** Cottrazm employed standard pre-processing for Morph, including log-normalization with a size factor 10,000 for each cell, and expression values z-score transformation for each gene across all spots. After running PCA dimensional reduction, KNN algorithm of *Seurat* package (v4.1.0)[81] was applied to cluster spatial spots. And the UMAP algorithm of *Seurat* package (v4.1.0) was applied to visualize cell subtypes.

A series of immune-related signatures, including pan-immune markers (PTPRC), pan-T cell markers (CD2, CD3D, CD3E, CD3G, CD5, CD7)[82] and B cell markers (CD79A, MS4A1, CD19)[83] were used to score spot. The average value of these features in Morph were denoted as the normal tissue expression score (NormalScore) of each spot. Based on the clustering results, Cottrazm selected the CNV in this cluster with highest median NormalScore was defined as CNV reference. Then, the pyramidal smoothing algorithm of package *Infercnv*[15] (v1.8.1) was

employed to obtain preliminary infercnv object based on either assay Spatial or Morph, Hidden Markov Model (HMM) was used to assess CNV level for remained spots. To more accurately classify spatial spots and distinguish malignant spots from non-malignant spots, Cottrazm employed hierarchical clustering based on tree partitioning in package *InferNCV* with random trees method to divide all spots into 8 clusters. Reference spots were labeled as "Normal". According to the gene with no CNV variation was scored 3, the gene with CNV amplification was scored greater than 3 and the gene with CNV deletion was scores less than 3, for gene_j of spot_i, its CNV score was denoted as $cs_{i,j}$, the CNV score of spot_i is denoted as $CS_i$, it is defined as follows:

$$CS_i = \sum_j |cs_{i,j} - 3| \tag{1}$$

The CNV scores of each spot were added to clustered Seurat object. In combination with the HE staining, CNVLabels with high median CNV scores (usually 2–4) were included in "MalLabels", and spots within these labels were initially defined as core spots of malignant spots.

Based on the clustering result, if more than half of spots in the cluster identified as MalLable, this cluster will be defined as malignant cluster. Cottrazm calculated the coordinates of centroid of malignant cluster $k$ ($Ci_{\mathrm{Mal},k}$) and normal cluster ($Ci_{\mathrm{normal}}$) based on UMAP embeddings. Then Cottrazm calculated the Euclidian distance from spot $i$ in malignant cluster $k$ to $Ci_{\mathrm{Mal},k}$ denoted as radius to $Ci_{\mathrm{Mal}}$ ($rt_{i,k}$) and to $Ci_{\mathrm{normal}}$ denoted as radius to $Ci_{\mathrm{normal}}$ ($rn_{i,k}$). The label of spot $i$ ($nbr_i$) was defined according to the following rule:

$$nbr_{i \in k} = \begin{cases} \text{Mal, if } rt_{i,k} < \frac{1}{3} \times rn_{i,k} \\ \text{unlabeled, if } rt_{i,k} \geq \frac{1}{3} \times rn_{i,k} \end{cases} \tag{2}$$

**Find neighbor spots of tumor core.** Cottrazm arrange spatial spots on hexagonal lattices and applied a natural way to define the neighboring spots. Briefly, utilize the spatial information, along each axis, using the image pixel coordinates and corresponding array coordinates of each spot to fit a liner model. Cottrazm then added the distances along each axis and multiplied this by a scaled factor to get the Manhattan distance, denoted as radius ($r$), the maximum distance between neighboring spots. Next, for any two spots in spatial (spot $i$ and spot $j$), the Manhattan distance between them ($pdist_{i,j}$) was calculated using the image pixel coordinate. When $pdist_{i,j} \leq r$, these two spots are considered neighbors, otherwise they are not neighbors.

**Calculating UMAP distance to tumor centroid in hexagon system.** Firstly, Cottrazm described malignant spots neighbored by non-malignant spots as "lonely malignant spots" (Mall). For system composed of $Mall_i$ and its neighbor spots ($nbr_i$), Cottrazm took the centroid of malignant cluster $k$ to which $Mall_i$ belongs as $Ci_{\mathrm{Mal},k}$ of the system and the centroid of the normal cluster as $Ci_{\mathrm{normal}}$ for the system. For neighbor spot $j$ of lonely malignant spots $i$ ($nbr_{i,j}$), Cottrazm calculated its Euclidean distances to $Ci_{\mathrm{Mal},k}$ and $Ci_{\mathrm{normal}}$ denoted as $rt_{i,j}$ and $rn_{i,j}$ respectively. Then, Cottrazm extrapolated the neighbor spot as malignant spot (Mal) or boundary spot (Bdy) based on the comparison of $rt_{i,j}$ and $rn_{i,j}$:

$$nbr_{i,j} = \begin{cases} \text{Mal, if } rt_{i,j} < \frac{1}{3} \times rn_{i,j} \\ \text{Bdy, if } rt_{i,j} \geq \frac{1}{3} \times rn_{i,j} \end{cases} \tag{3}$$

Next, in the first round of boundary extrapolation, the optimized malignant spots and the neighbor spots of lonely malignant spots considered as tumor spot were regarded as new malignant spots (MalSpotsN), in which case, all malignant spots have at least one neighboring malignant spot. New Seurat object (TumorSTn, n means

the round of extrapolation) composed of malignant spots and their neighbor spots, boundary spots, and normal spots was subset from original Seurat object with the labels (Mal, Bdy) of spots. For malignant spots $i$ (Mal$_i$) in MalSpotsN and the system $i$ composed of Mal$_i$ and its neighbor spots ($nbr_i$), Cottrazm firstly calculated the coordinate of malignant centroid in system $i$, denoted as $Ci_{\mathrm{Mal},i}$, then obtained the maximum Euclidean distance from malignant spot in the system to $Ci_{\mathrm{Mal},i}$, denoted as $r_{\mathrm{Mal},i}$.

If there were less than two spots in system $i$ labeled as Bdy, then for the unlabeled spot $j$ in system $i$ ($nbr_{i,j}$), Cottrazm calculated its Euclidean distance to $Ci_{\mathrm{Mal},i}$ and denoted as $p_{i,j}$. The label of $nbr_{i,j}$ was extrapolated as ($k$ was the scale factor):

$$nbr_{i,j} = \begin{cases} \mathrm{Mal}, \text{ if } p_{i,j} < k \times r_{\mathrm{Mal},i} \\ \mathrm{Bdy}, \text{ if } p_{i,j} \geq k \times r_{\mathrm{Mal},i} \end{cases} \quad (4)$$

If the number of spots in system $i$ labeled as Bdy is greater or equal to two, Cottrazm calculated the coordinates of the centroid of Bdy spots and Mal spots in system $i$ respectively and denoted as $Ci_{\mathrm{Bdy},i}$ and $Ci_{\mathrm{Mal},i}$. Then Cottrazm denoted the maximum distance from Bdy spots to $Ci_{\mathrm{Bdy},i}$ as $r_{\mathrm{Bdy},i}$ and the maximum distance from Mal spots to $Ci_{\mathrm{Mal},i}$ as $r_{\mathrm{Mal},i}$. For unlabeled spot $j$ in system $i$ ($nbr_{i,j}$), its Euclidean distance to $Ci_{\mathrm{Mal},i}$ ($p1_{i,j}$) and $Ci_{\mathrm{Bdy},i}$ ($p2_{i,j}$) were calculated. The label of $nbr_{i,j}$ was extrapolated as ($k$ was the scale factor):

$$nbr_{i,j} = \begin{cases} \mathrm{Mal}, \text{ if } p1_{i,j} < k \times r_{\mathrm{Mal},i} \text{ and } p2_{i,j} > r_{\mathrm{Bdy},i} \\ \mathrm{Bdy}, \text{ if } p1_{i,j} \geq k \times r_{\mathrm{Mal},r} \text{ or } p2_{i,j} \leq r_{\mathrm{Bdy},i} \end{cases} \quad (5)$$

If a spot was the neighbor of multiple Mal spots simultaneously, once it was labeled as Mal in any Mal spot system, this spot will be labeled as Mal; otherwise, this spot will be labeled as Bdy. After a round of extrapolation, Cottrazm added the new label of all neighbors of Mal spots to TumorSTn, the newly defined Mal spots will be denoted as MalSpotsN and entered the next round of boundary extrapolation.

Until the number of MalSpotsN is less than 3, or the number of neighbor spots of all the MalSpotsN is less than 3, or the times of extrapolation is more than 5, the boundary extrapolation will stop, and the round of extrapolation of general ST data can be 2–5 times. After boundary extrapolation was completed, the remained spots are label as non-malignant spots (nMal), which are neither Mal spots nor Bdy spots which belong to the outer tumor tissue.

### Tumor-specific signature score is relevance to the boundary accuracy

To access the tumor-specific score, we applied the 10–15 markers (Supplementary Table 2) mentioned in papers of CRC[3,31–33], HCC[11,34,35], BRCA[36–38], OV[39,40], ICC[35,41,42], and ccRCC[43–45] as tumor-specific signatures separately. Signature score was added to metadata of ST dataset with "AddModulScore" function with default parameters in Seurat. Spatial feature expression plots were generated with the "SpatialFeaturePlot" function in *Seurat* package.

### *Cottrazm-SpatialDecon*: Deconvolution of spatial spots
**scRNA-seq data preprocessing.** At the single-cell level, we need to generate a signature expression matrix (sig_exp) based on scRNA-seq data from the corresponding tissue. Briefly, differentially expressed (DE) genes were obtained for each of the identified cell type in scRNA-seq data using a Wilcoxon Rank Sum test from *Seurat* package "FindAllMarkers" function (only.pos = T, logfc_threshold = 0.25), the DE genes for each cell type were denoted as clustermarkers_list. Then Cottrazm calculated the mean expression of clustermarkers_list in each cell type and retuned a matrix sig_exp with genes as rows and cell types as columns.

**Selecting cell types in topics.** For deconvolution of spatial spots, Cottrazm mainly extended SpatialDWLS and modified the way generated the binary matrix for PAGE analysis and the selection of cell types entering deconvolution to make it more suitable for tumor data.

**Generating signature score matrix, enrichment score matrix, and topic.** 1) Signature score matrix: Cottrazm used the top 25 specifically expressed genes of each cell type in scRNA-seq reference to calculate the signature score of each cell type in each spot based on their average expression. 2) Enrichment score matrix: the binary matrix (enrich_matrix) is generated according to whether gene $j$ in clustermarkers_list of cell type $k$, the value of gene $j$ in the column of cell type k was 1, otherwise was 0. With enrich_matrix as input file, PAGE method is performed to generate an enrichment score matrix consisting of cell types as columns and spots as rows. 3) Topics vector is generated by the above KNN clustering result and the spot location information, including Mal, Bdy, and nMal.

**Determining cell types for each topic based on enrichment and signature score matrix.** Calculating cutoff values of signature score and enrichment score in each cell type, Cottrazm took the upper quartile of the signature score denoted as $cutoff_{\mathrm{sig},k}$ and the upper quartile of the enrichment score witch greater than zero denoted as $cutoff_{\mathrm{enrich},k}$. Then, Cottrazm selects the median signature score of cell type $k$ in all spots of topic $t$ is greater than $cutoff_{\mathrm{sig},k}$ as candidate cell type ($ct_{\mathrm{sig},t}$), and selects the top two cell types with max $enrich_{t,k}$ was greater than $cutoff_{\mathrm{enrich},k}$ as candidate cell types ($ct_{\mathrm{enrich},t}$). Determining candidate cell types of topics ($ct_t$) as follows. The number of cell types in ct$_{\mathrm{sig}}$ was denoted as n.

$$ct_t = \begin{cases} ct_{\mathrm{sig},t} \cup ct_{\mathrm{enrich},t}, \text{ if } n \geq 2 \\ ct_{\mathrm{sig},t} \cup \max ct_{\mathrm{enrich},t}, \text{ if } n = 1 \\ ct_{\mathrm{enrich},t}, \text{ if } n = 0 \end{cases} \quad (6)$$

To avoid malignant spots with specific signature affecting other cell types, Cottrazm took the union of the cell types with the 3 highest scores of max $enrich_{t,k}$ minus $cutoff_{\mathrm{enrich},k}$ and original $ct_t$ as final candidate $ct_t$. Since spots containing the fibrous tissue are difficult to permeate completely, these spots have low UMI count and tend to have non-significant signature scores and enrichment scores. In this situation, we will add stromal cells to $ct_t$ with stomal feature and median nCount_Spatial lower than 5000.

**Infering cell type composition of spots by DWLS method.** A weighted least squares approach was used to infer the composition of cell types. The weights were chosen to minimize the overall relative error rate. And the damping constant d determined by a cross-validation procedure was used to improve numerical stability. Cottrazm employed the same weights and $d$ on spots in the same topic to reduce technical variation. Then another round of deconvolution was performed after filtering the low proportion cell types.

**Evaluating Cottrazm's performance on deconvolution of spatial spots using simulated ST data**
**Simulating ST data from CRC scRNA-seq.** To simulate spot-like transcriptomic data, we generated simulated ST data from CRC scRNA-seq data based on the spatial location that we dissected on *Cottrazm-BoundaryDefine* and cell-type distribution patterns in TME. In the simulated ST dataset, each spot consists of a mixture of 6–10 cells randomly picked from scRNA-seq data according to the information provided by 10X Genomics and followed the following rules to maximize the simulation of the true situation of TME. We divided simulated ST data into malignant region with 1000 spots, tumor boundary region with 500 spots, non-malignant region with 1000 spots, and TLS regions with 250 spots. The cell composition of spots in each region

based on random selection of the results obtained from *Cottrazm-SpatialDecon*. Summarize UMI counts of selected cells into transcriptome profiles to represent the spot's expression profile. If the simulating mixture had >50,000 UMI counts, we randomly down sampled it to 50,000 UMI counts to better simulate biological situation of spatial transcriptomics data. Since the selected cells are derived from scRNA-seq analysis and their cell type identities are known, the resulting cell composition be considered as the gold standard for evaluating deconvolution performance.

**Evaluating performance of Cottrazm-SpatialDecon.** To address how well the *Cottrazm-SpatialDecon* performed, we assessed if the predicted proportions accurately represented the true composition through *accuracy*, percentage of correctly classified cell types, *F1 score*, combined sensitivity and precision, indicating how-corrected and how-well we are at identifying cell-types present, respectively, and *specificity*, predicting cell-types at absent spot. Furthermore, we also used other two metrics, correlation on spot-level and cell type-level, JSD distance, to evaluate the deconvolution performance. For each cell type, Spearman's correlation between the proportions of predicted and true cell types to assess the model performance of distinguishing different cell types. For each spot, Spearman's correlation between cell composition of prediction and the ground truth to assess the deconvolution accuracy in each spot. JSD distance metric was used to measure the similarity between predicted and true cell type proportions in each spot.

**Benchmarking different deconvolution tools.** We benchmarked *Cottrazm-SpatialDecon* against other cell-type deconvolution tools of spatial transcriptomics, including RCTD under spacexr v2.0.0, SpatialDWLS[49], STRIDE[50] v0.0.2a, and SPOTlight[51] v0.99.11, as well as the ones for bulk RNA- seq, including CIBERSORTx[52] under Docker Container and MuSiC[53] v0.2.0. All tools were run with default settings specified in their documentation. Performances of these tools were assessed as described in the "*Evaluating performance*" section.

***Cottrazm-SpatialRecon*: Reconstruction of spatial gene expression matrix for sub-spots.** Cottrazm reconstructed the gene expression matrix of infiltrated cells as sub-spots at almost the single-cell level according to an algorithm based on the feature weight of cell types, spot deconvolution result, and ST feature expression. In the signature expression matrix (sig_exp) of single cell level we obtained above, the mean expression of feature $j$ in cell type $k$ was denoted as $expr_{j,k}$, the contribution of feature $j$ to the annotation of cell type $k$ was denoted as $w_{j,k}$, so we have:

$$w_{j,k} = \frac{expr_{j,k}}{\sum_{k} expr_{j,k}} \qquad (7)$$

In the deconvolution result, the proportion of cell type $k$ in spot $i$ was denoted as $d_{i,k}$. In the expression matrix of ST data, the expression of feature $j$ in spot $i$ was denoted as $f_{i,j}$. So, the expression of feature $j$ in sub-spot $k$ split from spot $i$ ($f_{i,j,k}$) is defined as follow:

$$f_{i,j,k} = \frac{w_{j,k} \times d_{i,k}}{\sum_{k} w_{j,k} \times d_{i,k}} f_{i,j} \qquad (8)$$

Then Cottrazm synthesized the reconstructed sub-spot matrix of ST data according to the feature of each sub-spot matrix after splitting and generated the reconstructed TME Seurat object with the "CreateSeuratObject" function. The cell types of deconvolution were added to reconstructed Seurat object and the original spot id were added as orig.ident, and the result of boundary extrapolation was added as Location.

## Reduction and clustering of reconstructed spatial TME data

For reconstructed Seurat object of spatial sub-spot, we filtered according to certain condition (nFeature_RNA > 400 & 400 <nCount_RNA < 50,000) for downstream analysis. The filtered Seurat object were normalized and scaled, the top 2000 most variable genes were found by using "FindVariableFeatures" function and then used for PCA (npcs = 30). We used "FindNeighbors" in *Seurat* to get nearest neighbors for graph clustering based on top 20PCs, and "FindCluster" (resolution = 1) to obtain cell subtypes. And the UMAP algorithm was applied to visualize cell subtypes.

## Evaluating performance of *Cottrazm-SpatialRecon*

Base on mixture of cell types created by "*Simulating ST data from CRC scRNA-seq*" section, we used *Cottrazm-SpatialRecon* to reconstruct the cell type specific gene expression profiles at sub-spot level. Then, we merge gene expression matrix from simulating ST data and Cottrazm prediction and utilized RunHarmony method in R package *harmony* to remove the batch effects. FindNeighbors and FindCluster in *Seurat* were used to cluster cell subtypes, and UMAP was used to visualize the distribution of cell types between prediction and the truth. We further performed Spearman's correlation to estimate the concordance between gene expression profile of prediction and the ground truth in each cell types.

## Dependencies for Cottrazm

Other dependencies of Cottrazm including R packages *magrittr* (v2.0.3), *dplyr* (v1.0.8), *Matrix* (v1.4.0), *ggplot2* (v3.3.5), *stringr* (v1.4.0), *RColorBrewer* (v1.1.3), *patchwork* (v1.1.1), *ggtree*[84] (v3.0.4), *BioGenetics* (v0.38.0), *readr* (v2.0.0), *rtracklayer* (v1.52.1), *phylogram* (v2.1.0), *utils* (v4.1.0), *dendextend*[85] (v1.15.1), *assertthat* (v0.2.1), *reticulate* (v1.22), *openxlsx* (v4.2.4), *scatterpie* (v0.1.7), *cowplot* (v1.1.1), *stats* (v4.1.0), *quadprog* (v1.5.8), *data*.table (v1.14.2), *Rfast*[86] (v2.0.6), *ggrepel* (v0.9.1), *tibble* (v3.1.6), *clusterProfiler*[87] (v4.05.), *org.Hs.eg.db* (v3.13.0); and python modules *argparse* (≥1.1), *scanpy*[88] (≥1.7.1), *numpy* (≥1.19.5), *pandas* (≥1.3.4).

## Data preprocess of spatial transcriptomics

Raw sequencing data of spatial transcriptomics were quality checked and mapped by SpaceRanger v1.1 Spatial transcriptome data were qualitatively controlled using parameters including total spots, median UMIs/spot, median genes/spot, median mitochondrial genes/spot. Spots used in the subsequent analysis were filtered for minimum detected gene count of 200 genes while genes expressed in fewer than 3 spots were removed. Normalization across spots was performed with the LogVMR function in *Seurat*.

## Analysis of differentially expressed genes

The function "FindAllMarkers" in *Seurat* was used to identify genes differentially expressed among clusters. The function "FindMarkers" in *Seurat* was used to find differentially expressed genes between two groups. The Welch t-test algorithm was used to identify p value of differentially expressed genes between boundary spots and other spots. The non-parametric Wilcoxon rank-sum test was used to obtain p-values for comparison, as well as the adjusted p-values for all genes in the dataset based on the Bonferroni correction.

## Projection signature scores of Macro-SPP1 and Fib-APSN to ST datasets

The top 100 differentially expressed genes of Macro-*SPP1* and Fib-*ASPN* separately derived from the reconstruction of CRC and HCC datasets was overlapped to obtain the signature score for Macro-*SPP1* and Fib-*ASPN*. Signature scores were added to metadata of ST dataset of BRCA, ccRCC and OV with "AddModulScore" function with default parameters in Seurat. Spatial feature expression plots were generated with the "SpatialFeaturePlot" function in *Seurat* package.

## Cell-cell interaction analysis

We applied R package *Cellchat* (v1.1.1)[89] to evaluate the interaction weights of boundary enriched subclusters of boundary spots and its neighbor spots, implemented with default parameters. Briefly, differentially expressed signaling genes ($p < 0.05$) were first identified across cell clusters in the reconstructed ST sub-spots dataset to infer specific cellular communications.

## Tumor boundary signature score is relevance to lymphocyte infiltration

To assess the signature score on the tumor immune exclude microenvironment (TIEM)-specific tumor boundary, immune excluded score (ieScore), we obtained the overlapping genes of the top 100 specifically expressed genes in the tumor boundary of the three CRC ST samples as the signature genes. We used GSVA[70] to estimate ieScore in each patients from TCGA CRC cohort and stratified the patients into ieScore-high and -low group based the median score, then compared the lymphocytes and CD8$^+$ T infiltration between these two groups. For patients' overall survival (OS) and progression-free survival (PFS) data analysis, the Kaplan–Meier method and log-rank test were used to detect differences in the survival curves between groups. The Cox proportional hazard model was used to assess the relative risk by setting the expression of Schwann cell abundance as a single covariate.

## Reporting summary

Further information on research design is available in the Nature Portfolio Reporting Summary linked to this article.

## Data availability

This study made use of publicly available spatial transcriptomics, colorectal cancer dataset[3] HRA000979 and hepatocellular carcinoma and intrahepatic cholangiocarcinoma dataset[11] HRA000437 were obtained from Genome Sequence Archive (GSA) under restricted access. Access can be requested through the GSA access committee. Breast cancer (BRCA1, https://www.10xgenomics.com/resources/datasets/human-breast-cancer-block-a-section-1-1-standard-1-1-0; BRCA2, https://www.10xgenomics.com/resources/datasets/invasive-ductal-carcinoma-stained-with-fluorescent-cd-3-antibody-1-standard-1-2-0) and ovarian cancer (https://www.10xgenomics.com/resources/datasets/human-ovarian-cancer-whole-transcriptome-analysis-stains-dapi-anti-pan-ck-anti-cd-45-1-standard-1-2-0) were obtained from 10X Genomic website. Renal cell carcinoma GSE175540 [https://www.ncbi.nlm.nih.gov/geo/query/acc.cgi?acc=GSE175540][30] and squamous cell carcinoma GSE144240 [https://www.ncbi.nlm.nih.gov/geo/query/acc.cgi?acc=GSE144240][14] were obtained from Gene Expression Omnibus (GEO). The details of all samples with spatial transcriptomics used in this study provided in Supplementary Tables 1. For the scRNA-seq used in this study, including processed breast cancer data (https://lambrechtslab.sites.vib.be/en/single-cell)[90], processed ovarian cancer data (https://lambrechtslab.sites.vib.be/en/high-grade-serous-tubo-ovarian-cancer-refined-single-cell-rna-sequencing-specific-cell-subtypes)[91], and colorectal cancer[54] GSE132465, GSE132257, and GSE144735. The bulk RNA-seq data for colorectal cancer from the TCGA data portal (http://gdac.broadinstitute.org/), which based on fragments per kilobase of exon model per million reads mapped (FPKM).The simulated datasets generated in this study were deposited at Mendeley Data (https://data.mendeley.com/datasets/m5s3zjrs6j/1). The lymphocytes and CD8$^+$ T infiltration of TCGA CRC cohort were obtained from Thorsson et al.[92] (https://gdc.cancer.gov/about-data/publications/panimmune). Source data are provided with this paper.

## Code availability

The Cottrazm software package depends on R language and source code have been deposited at https://github.com/Yelab2020/

Cottrazm[93] and achieved at https://doi.org/10.5281/zenodo.7519134[94]. The package vignette to reproduce the process of Cottrazm is available in the same website. Other scripts used to reproduce analyses for plotting figure are available from the author upon request.

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

## Acknowledgements

This work was supported by grants from the National key research and development program (2022YFC2504700 to Y.Y.), National Natural Science Foundation of China (81971487 and 82203189 to Y.Y., 32100730 to S.L.), Shanghai Science and Technology Commission (20JC1410100 to Q.C.), Natural Science Foundation of Shanghai (20ZR1472900 to Y.Y.), Shanghai Jiao Tong University 2030 Initiative (WH510363001-4 to Y.Y.), and National Postdoctoral Program for Innovative Talent (BX2021188 to S.L.). We thank the support from sequence core at Shanghai Institute of Immunology.

## Author contributions

Y.Y. conceived and supervised the project. Z.X., L.H., and Y.Y. developed and implemented Cottrazm. X.D., Z.X., and Y.Y. conducted all analyses. Y.Z., D.Z., and B.Z. performed pathological annotation, S.L., B.S., G.C., J.Z., X.Z., L.H., and Y.Y. involved in interpretation, Z.X., L.H., and Y.Y. wrote the manuscript.

## Competing interests

The authors declare no competing interests.
