## [Peer review file · Nature Communications]

REVIEWER COMMENTS

Reviewer #1 (Remarks to the Author): Expert in spatial transcriptomics, scRNA-seq, tumour microenvironment, and cancer genomics

The manuscript describes a new computational method for the analysis of spatial transcriptomics datasets through integration with the H&E images, focusing on the tumor-microenvironment boundary. The tool, called Cottrazm, presents a novel approach for study of the border between malignant and non-malignant cells, and the authors are commended for developing this useful resource. We have the following specific comments regarding the manuscript.

1. For the find boundary function, it would be useful to have pathologist annotation in order to validate the results from the tool.
2. Most tumor regions have infiltrating macrophages, fibroblasts, lymphocytes etc. How does the tool deal with this? Does the tool assume that the malignant core is made up of purely malignant cells? This is an important caveat that has not been addressed.
3. Fig 4 - what is a subspot?
4. Fig 5, it would be important to also show validation using an external/orthogonal method that the differential localization is real. For eg, FISH or immunostaining to show differential localization of the different subspots.
5. Since the focus of the tool is based on the boundary, it would be useful to show the utility of the tool in identifying new biology. The differential localization of different macrophage or fibroblast subtype is extremely interesting but the novelty of the finding would be better emphasized if there is a biological impact observed.
6. With respect to the tumor boundary signature, what is the difference between the boundary Score high and low patients? In vivo all tumors would have a boundary, hence it would be useful to describe potentially a type of boundary instead of boundary high or low.

Reviewer #2 (Remarks to the Author): Expert in tertiary lymphoid structures, tumour immunology, spatial transcriptomics, and computational genomics

Xun et al. present in this manuscript a method called Cottrazm that facilitates the analysis of spatial transcriptomics of tumors, by allowing to annotate the tumor zone, boundary and non-malignant areas,

deconvolute each spot's signal to evaluate the cell type composition, and decipher cell-type specific gene expression profiles.

While parts of this workflow have already been covered by other approaches, other parts are novel, and the authors show that they outperform existing deconvolution methods. The field of spatial transcriptomics for cancer is moving fast, and it is likely that this method would be of significant interest for the community.

Overall, the novel approach is interesting, well thought and described adequately. The manuscript is well written and organised. However, I have some interrogations on crucial aspects that would need to be addressed in a revised version. I apologize for the long list of points below, but I believe addressing them would strengthen this promising manuscript.

Major points:

1) To annotate the different regions, the authors use the Euclidian distances between spots on a UMAP projection. This has the following pitfalls:

a. UMAP is not a linear process and the distances on the map are only partially interpretable. Relying on them quantitatively is usually strongly discouraged.

b. UMAP is a stochastic process, and running the algorithm with a different seed would give different maps from the same origin dataset, and therefore different Euclidian distances.

With these issues in mind, I believe that Euclidian UMAP distances cannot be used as a robust indicator, and I would strongly encourage the authors to use a different metric. For instance, they could consider Euclidian distance on the 50 PC projection, since Euclidian distance easily generalises to any number of dimensions. Maybe the metric could also be weighted to favor the first PCs that bear most of the variance. Since PCA is a linear and deterministic process, such distances would be interpretable and not dependant on heuristic process.

2) I am not sure why the authors chose this method to score normal tissue. In the manuscript, they rely on the expression of immune genes. I have the following concerns with this:

a. There can be presence of immune cells in the tumor core as well, so defining the non malignant area by their presence is a biased approach, and would result in a wrongly estimated near absence of immune cells in spots considered to be malignant.

b. Even is this approach was valid, the list of genes is surprising: CCR7 can be expressed in most immune cels, but also by some tumor cells. AICDA is only expressed by germinal center B cells. CXCR6 and FOXP3 are related to some T cell subsets. It is surprising to see them considered at the same level as pan-immune markers or pan-T cell markers. There are additionally no markers of myeloid cells, such as macrophages (the most abundant immune cells in many malignancy!), dendritic cells, monocytes, neutrophils, ...

c. Why label non malignant tissue based on immune signatures? Why not include genes related with normal stroma?

3) Cottrazm's annotation of tumor region needs to be adequately validated against what could be considered as an appropriate gold standard. I would consider pathologist's annotations of tumour core, invasive margin and adjacent tissue to be gold standard. The authors should consider collaborating with one or several pathologist to evaluate the agreement between Cottrazm and pathologists, paying attention to separate their datasets into training and validation.

4) Spot deconvolution should be validated against other technics as well, such as immunohistochemistry, not only against deconvolution methods. There is a public dataset of Visium and CD3 (T cells) staining on invasive ductal carcinoma that the authors could consider using to compare their estimates with immunostaining results: https://support.10xgenomics.com/spatial-gene-expression/datasets/1.2.0/V1_Human_Invasive_Ductal_Carcinoma

5) In the benchmark against other deconvolution approaches, some methods are missing, including Stereoscope (<https://doi.org/10.1038/s42003-020-01247-y>) and Cell2Location (<https://doi.org/10.1038/s41587-021-01139-4>).

6) The use of Spearman correlation seems inadequate. A linear relationship between truth and prediction is expected, so Pearson correlation would be a much more appropriate metric.

7) To my understanding, the results presented in Fig. 3e-h are based on the same scRNA-seq dataset used for training (generation of the simulated data) and validation. This is biased, and 3f's "truth" should come from an independent validation dataset that is not used to generate the simulated Visium data.

8) The simulated dataset was limited to a maximum of 10 cells/spot. Where does this limitation come from? In dense areas, such as TLS, the number of spots can often be much higher than 10, probably around ~30 cell/spot. The authors should consider simulating some spots with higher cell type densities.

Minor points:

1) Some of the appropriate literature in the field is not cited.

a. The authors should consider discussing their findings in light of the reports of the impact of TGFb leading to immune-excluding tumors unresponsive to ICB (<https://doi.org/10.1038/nature25501>).

b. The deleterious impact of SPP1+ macrophages has also already been reported (see <https://doi.org/10.1038/s41571-022-00620-6> for a review).

c. Ref 53 (Cabrita et al.) was published alongside 2 other articles (<https://doi.org/10.1038/s41586-019-1906-8> and <https://doi.org/10.1038/s41586-019-1922-8>) that extend the results to other cancer types, and this would further the discussion.

d. CellChat is used, but not cited (<https://doi.org/10.1038/s41467-021-21246-9>). Please consider citing it so that the authors can get credit for their work.

2) The authors could consider adding a third UMAP on Fig. 4c that would show the region of origin (Mal/Bdy/nMal).

3) How was the TCGA dataset separated between High and Low boundary score? Was it median cut-off? This needs to be specified.

4) Cottrazm seems to be only for 10x Genomics Visium data, not for other spatial transcriptomics modalities (e.g. SlideSeq2, Nanostring GeoMx, ...). This needs to be made clear in the manuscript.

5) The language in which Cottrazm is available (R?) should be specified in the text.

Reviewer #3 (Remarks to the Author): Expert in spatial transcriptomics, scRNA-seq, and bioinformatics

Xun and colleagues present Cottrazm, a bespoke method for identifying malignant, boundary and non-malignant spots in spatial transcriptomics data. This method works by performing InferCNV on spatially resolved spots to identify potentially malignant spots, then expanding around the border of such malignant spots to assign boundary spots. The authors demonstrate the utility of Cottrazm by applying it to samples across several cancer types, and identify a common pattern of expression of expression, likely driven by localisation of specific subtypes of macrophage and fibroblast, present in the border regions across tumours. I find the work is innovative and promising for further understanding the tumour microenvironment. The manuscript is clear and well-motivated, and the software is available online. I have the following comments regarding the manuscript:

- It's unclear what motivates the various choices within the Cottrazm-SpatialDecon method, e.g. what motivates the choice of PAGE for the gene set enrichment analysis, and DWLS as opposed to say typical least squares regression?

- How relevant is the size of the spots for determining? e.g. if the spot data was a higher resolution would you be extracting different boundary regions? seems to be the case given the parametrisation on neighbouring spots. As it stands it's unclear whether such method is applicable to a technology with different spot size/resolution as the boundary could be thinner or thicker depending on the technology (and not the biology). The authors should discuss the relative merits to determining boundary by the nearest neighbours versus an absolute distance (e.g. in microns).

- The point above could be tested by artificially combining spots to a lower resolution and examining the concordance of malignant/boundary/non-malignant predictions.

- How applicable is InferCNV to non-single-cell data as used here (per spot)? Could this lead to artefacts in the InferCNV output due to input of multiple cells, possibly mixtures of malignant and non-malignant cells? Additionally, it may be that not all cancers result in such large copy-number changes but rather have a large SNV burden (e.g. melanoma), this limitation should be discussed in the text.

- What ground truth is present? e.g. Figure 2a appears like these are the outputs of Cottrazm, are there expert/pathologist annotations of these slides to refer to?

- What is the false negative rate of Cottrazm? i.e. will it always assign spots to "Malignant"? This can be tested by performing Cottrazm on an adjacent normal tissue.

- The manuscript states "All scripts used to reproduce all the analyses are also available at the same website." but I can't seem to find this on the Github page. Please specifically link to the scripts and/or make these available on the Github.

- No plots visible in the package vignette as all code is commented out, this should be amended to display the expected output in the vignette.

Minor

- Figure 1 schematic of tabular representation of "Spots" and "HE stain" doesn't quite make sense - the dimensionality is very unlikely to be equal to the genes in the "Spots" "Feature" matrix, and it's unclear what the rows actually represent, the HE stain looks like both rows and columns are spots and the colour of the cells themselves are the HE stain values. In addition, it would be useful to include a colour legend for this schematic too, what do the light to dark pink colours represent in the "Spots" "Morph adjusted feature" matrix?

- typo in Figure 1 "Neighbors" and "Figure"

- Figure 2 b boxes overlaid on violin plots should either be reordered or some transparency used, as it masks the violin's distribution

Point-by point Responses to Reviewers' Comments:

Reviewer #1 (Remarks to the Author): Expert in spatial transcriptomics, scRNA-seq, tumour microenvironment, and cancer genomics

The manuscript describes a new computational method for the analysis of spatial transcriptomics datasets through integration with the H&E images, focusing on the tumor-microenvironment boundary. The tool, called Cottrazm, presents a novel approach for study of the border between malignant and non-malignant cells, and the authors are commended for developing this useful resource. We have the following specific comments regarding the manuscript.

Response: We thank the reviewer for overall positive comments that highlight the significance of our study.

1. For the find boundary function, it would be useful to have pathologist annotation in order to validate the results from the tool.

Response: We thank the reviewer's constructive suggestion. We obtained a new spatial transcriptomics of squamous cell carcinoma (SCC) performed by 10X Genomics platform and the H&E staining with leading edge annotated by pathologist in previous study¹. Then we asked the professional pathologist from Ruijin Hospital, China to help us annotate the tumor boundary in representative H&E staining which used in our original study, including breast cancers (BRCA), colorectal cancer (CRC), and hepatocellular carcinoma (HCC) (**Additional Figure [Fig. A]1a**). Then, we imported the outlined tumor boundary layer into R and converted into an unordered set of points (pixel coordinates), each point defining a position on the tumor boundary layer. Further, we calculated pairwise distances between spots of ST and tumor boundary outline by k-Nearest Neighbor algorithm (kNN, dbscan R package) to find the minimum distance from a spot to the closest outline (**Fig. A1b-d**). We considered the tumor boundary spot predicted by our tool Cottrazm is match to the location of pathologist-annotated tumor boundary outline when their distance is less than one spot size. Our result showed highly consistency between predicted tumor boundary spot and pathologist-annotated tumor boundary outline, ranging from 91.30% in SCC to 99.73% in CRC (**Fig.A1e**), suggesting the great performance of our tool on the delineation of tumor boundary. We added **Fig. A1** as **Fig. 2e-h**, **Supplementary Fig. 1f**, and **Supplementary Fig. 2b-d** and the result of concordance between pathologist annotation and prediction of Cottrazm in revised manuscript (page 8-9 line 185-201)

Fig. A1. The concordance of pathologist annotation and prediction of Cottrazm. (a) HE stained images with pathologist annotated tumor boundary of squamous cell carcinoma (SCC), breast cancer (BRCA), colorectal cancer (CRC), and hepatocellular carcinoma (HCC). (b) Tissue slides were annotated by malignant spots (Mal, red), boundary spots (Bdy, blue), and non-malignant spots (nMal, orange), including SCC, BRCA, CRC, and HCC. (c) Tumor boundary annotated by pathologist (black) and boundary spots (b) and boundary spots. (d) Tumor boundary annotated by Cottrazm (black). (e) Stacked bar chart showing concordance for SCC, HCC3, CRC1, and BRCA.

annotated by Cottrazm (Blue). **(d)** Line segments of the shortest distance from boundary spots annotated by Cottrazm to the pathologist's boundary. **(e)** Bar plot showing the proportion of boundary spots which distance to pathologist's boundary is less than or equal to one spot.

2. Most tumor regions have infiltrating macrophages, fibroblasts, lymphocytes etc. How does the tool deal with this? Does the tool assume that the malignant core is made up of purely malignant cells? This is an important caveat that has not been addressed.

Response: Thank you for the reviewer's comment. Indeed, in most malignant regions may have infiltrating macrophages, fibroblasts, lymphocytes, etc. In our tool, we assume that the malignant core consisted of the most malignant cells than other regions, which contributes to most copy number alterations. Here, we compared the CNV scores of the malignant core region with other regions and found that the malignant core had significantly higher CNV scores than other regions across multiple cancer types, including CRC, BRCA, HCC, and OV (**Fig. A2a**). Then, we validated the cellular composition of malignant core region. Although there are a small number of other cells (*e.g.*, macrophages, fibroblasts) in the malignant core area, most of them are composed of malignant cells in multiple cancer types (**Fig. A2b**). With this knowledge, we did not assume that the malignant core is made up of purely malignant cells. Instead, we assume the malignant core with highest CNV scores and composed of the highest proportion of malignant cells. We clarified this in revised manuscript (page 6, line 123-124)

Fig. A2. The CNV score and cellular composition of malignant regions. (a) Box and violin plot compared the CNV score of tumor core spots (red) and spots in other regions (blue) of CRC, BRCA, HCC, and OV. **(b)** Bar plots representing the proportions of the seven cell types predicted by Cottrazm in the spot of tumor core of CRC, BRCA, HCC, and OV.

3. Fig 4 - what is a subspot?

Response: We defined sub-spot as different cell types in each spot. We clarified this in the revised manuscript (page 12, line 299-300).

4. Fig 5, it would be important to also show validation using an external/orthogonal method that the differential localization is real. For eg, FISH or immunostaining to show differential localization of the different subspots.

Response: We obtained ST dataset from published studies, so unfortunately, we cannot obtain the same slide with FISH or immunostaining and ST data. To address the reviewer's concern, we obtained samples with immunostaining from human protein atlas (HPA)². We found immunostaining for PLVAP and ACKR1 in CRC samples from the same patient (id: 2096), and

consistent with the results predicted by Cottrazm, PLVAP staining was enriched at the tumor boundary, while ACKR1 was scattered in the stroma (**Fig. A3a**). Similar as this, in a stained sample from another CRC patient (id: 3264) we found the ASPN stain tended to enrich at tumor boundary, while POSTN, representative marker of another subtype of fibroblasts, tended to enrich at tumor stroma (**Fig. A3b**). Consistent results were observed in the sample of BRCA patients (id: 1939, **Fig. A3c**). These results suggested that Endo-PLVAP, Fib-ASPN tend to enrich in tumor boundary, while Endo-ACKR1 tended to enrich in stroma non-malignant region. Taken together, these data demonstrated that the differential localization is real, and we added **Fig. A3b-c** as **Supplementary Fig. 6g-h** and added this data in the revised manuscript (page 15, line 365-371).

Fig. A3. Differential distribution of cell subtype signatures performed by immunostaining. (a) HPA stain of PLVAP (upper), ACKR1 (bottom), of CRC (b-c) HPA stain of ASPN (upper), POSTN (bottom), of CRC (b) and BRCA (c).

5. Since the focus of the tool is based on the boundary, it would be useful to show the utility of the tool in identifying new biology. The differential localization of different macrophage or fibroblast subtype is extremely interesting but the novelty of the finding would be better emphasized if there is a biological impact observed.

Response: As the suggested by the reviewer, we compared functional enrichment between different macrophage or fibroblast subtypes. For the functional roles of macrophage subtypes, we found that Macro-*SPP1*, which localized along the boundary, mainly functionally enriched in cell migration and metabolic, while Macro-*FOLR2*, localized around TLS in the non-malignant region, mainly functionally enriched in antigen presentation (**Fig. A4a**).

As described in previous studies, the functional phenotypes of tumor associated macrophage in pan-cancer can be defined as dichotomous functional phenotypes, including angiogenesis and phagocytosis^{3,4}. Using the angiogenic and phagocytic signatures, we assessed the functional phenotypes of two macrophage subtypes with differential localization. We found angiogenesis signature genes preferentially expressed in Macro-*SPP1*, while phagocytosis signature genes enriched in Macro-*FOLR2* (Fig.A4b). In addition, Macro-*SPP1* showed significantly higher M2 score than Macro-*FOLR2*, suggesting the key roles of Macro-*SPP1* in the tumorigenesis (Fig. A4c).

For the function roles of fibroblast subtypes, we found that Fib-*ASP*, which tends to localize along the tumor boundary, is highly active in pathways that contribute to the formation of desmoplastic structures, including extracellular matrix assembly, collagen fibril organization, and collagen biosynthetic process (Fig. A4d), while Fib-*SFRP2* localized in the non-malignant region functionally enriched in collagen degradation and mesenchyme migration. We highlighted the utility of our tool in identify new biology in the revised manuscript. We added Fig. A4a-c as Supplementary Fig. 5f-h; Fig. A4d as Supplementary Fig. 6e and added this data in the revised manuscript (page 13, line 318 – 330; page 14-15, line 355-360).

Fig. A4. Biological function of different macrophage and fibroblast subtypes with different location. (a) Bar plots showing the differentially enriched Gene ontology (GO, left) and Kyoto Encyclopedia of Genes and Genomes (KEGG, right) terms of Macro-SPP1 and Macro-FOLR2. (b) Box and violin plots

showing the angiogenesis (left) and phagocytosis (right) scores of Macro-SPP1 and Macro-FOLR2 in reconstructed CRC data. **(c)** Box and violin plots showing the M1 (left) and M2 (right) scores of Macro-SPP1 and Macro-FOLR2 in reconstructed CRC data. **(d)** Bar plots showing the differentially enriched GO (left) and KEGG (right) terms of Fib-ASP1 and Fib-SFRP2 in reconstructed CRC samples.

6. With respect to the tumor boundary signature, what is the difference between the boundary Score high and low patients? In vivo all tumors would have a boundary, hence it would be useful to describe potentially a type of boundary instead of boundary high or low.

Response: We thank the reviewer's great suggestion. The tumor immune microenvironment of spatial transcriptomics slides in Fig.5 were defined as tumor immune excluded microenvironment (TIEM). We found there are specific boundary gene signature related to TIEM. To determine the characteristics of TIEM, we used the gene signature score on the TIEM-specific tumor boundary to assess the level of immune exclusion in tumor. We revised the boundary Score as "immune excluded score (ieScore)" in revised manuscript to avoid the confusion (page 17, line 418-421).

Reviewer #2 (Remarks to the Author): Expert in tertiary lymphoid structures, tumour immunology, spatial transcriptomics, and computational genomics

Xun et al. present in this manuscript a method called Cottrazm that facilitates the analysis of spatial transcriptomics of tumors, by allowing to annotate the tumor zone, boundary and non-malignant areas, deconvolute each spot's signal to evaluate the cell type composition, and decipher cell-type specific gene expression profiles. While parts of this workflow have already been covered by other approaches, other parts are novel, and the authors show that they outperform existing deconvolution methods. The field of spatial transcriptomics for cancer is moving fast, and it is likely that this method would be of significant interest for the community.

Overall, the novel approach is interesting, well thought and described adequately. The manuscript is well written and organised. However, I have some interrogations on crucial aspects that would need to be addressed in a revised version. I apologize for the long list of points below, but I believe addressing them would strengthen this promising manuscript.

Response: We are very pleased that the reviewer considers our study is interesting and well thought. We greatly appreciated the constructive suggestions from the reviewer. We have addressed all comments with the point-by-point responses as follows.

Major points:

- 1) To annotate the different regions, the authors use the Euclidian distances between spots on a UMAP projection. This has the following pitfalls:
 - a. UMAP is not a linear process and the distances on the map are only partially interpretable. Relying on them quantitatively is usually strongly discouraged.
 - b. UMAP is a stochastic process, and running the algorithm with a different seed would give different maps from the same origin dataset, and therefore different Euclidian distances. With these issues in mind, I believe that Euclidian UMAP distances cannot be used as a robust indicator, and I would strongly encourage the authors to use a different metric. For instance, they could consider Euclidian distance on the 50 PC projection, since Euclidian distance easily generalises to any number of dimensions. Maybe the metric could also be weighted to favor the first PCs that bear most of the variance. Since PCA is a linear and deterministic process, such distances would be interpretable and not dependant on heuristic process.

Response: Thank you for the reviewer's constructive suggestions. Although the UMAP is not a linear process and is a stochastic process, it can present a minimum distance between nearest neighbors in low-dimensional space. UMAP embeds the data into a Euclidean space by default (<https://search.r-project.org/CRAN/refmans/uwot/html/umap.html>, <https://github.com/lmcinnes/umap>), so using Euclidean distance makes sense. To validate the consistent Euclidean distance of two random spots among different seeds, we randomly selected 10 pairs of spatial spots in sample BRCA and calculated the Euclidean distance between them at 1-100 seeds. The Euclidian UMAP distances of two spots couldn't change when used different seeds (**Fig. A5a**). To assess the consistent result by the Euclidian UMAP distances, we set seeds range from 1 to 100 and predicted the tumor boundary. We found the tumor boundary spots are completely consistency among these 100 tests (**Fig. A5b-c**).

As suggested by the reviewer, we used weighted Euclidian distance of 10, 20, 30, 40, 50 PCs, and predicted the tumor boundary. Then we assessed concordance of pathologist annotation and predicted tumor boundary based on weighted Euclidian distance of PCs range from 10 to 50. Take the BRCA slide with ST as an example, the predicted tumor boundary differed among different PCs. When using 50 PCs, some spots belonging to Mal are not defined as Mal (marked by white boxes, black lines are the tumor boundary annotated by pathologist); when using 40 or 30 PCs, not only some spots belonging to Mal are not defined, but also some spots belonging to nMal are defined as Mal; When using 10 or 20 PCs, some spots belong to nMal are defined as Mal (**Fig. A5d**). Predicted tumor boundary spots based on weighted Euclidian distance of PCs match to pathologist-annotated tumor boundary outline range 64.29% in 50PCs to 75.24% in 30PC (**Fig. A5e**), while there are 94.46% spots based on UMAP distance overlap with pathologist-annotated spots. Taken together, we consider the Euclidean distances between spots on an UMAP projection is appropriate to assess the similarity of two spots.

Fig. A5. Boundary extrapolation results of different UMAP seeds and weighted PC projection. (a) Dot plot showing the Euclidian distance between ten randomly selected paired spots among 100 different seeds. **(b)** BRCA slides were annotated by Mal, Bdy, nMal with seeds range from 1 to 100. **(c)** Bar plot showing the proportion of boundary spots of different seeds consistent with original seed (888). **(d)** BRCA slides were annotated by Mal, Bdy, and nMal using different weighted PCs (50, 40, 30, 20, 10). The black line indicates the tumor boundary annotated by pathologist. The white square boxes display the inconsistent region of tumor boundary prediction based on PCs and tumor boundary annotated by pathologist. **(e)** Bar

plot showing the proportion of boundary spots which distance to pathologist's boundary is less than or equal to one spot.

2) I am not sure why the authors chose this method to score normal tissue. In the manuscript, they rely on the expression of immune genes. I have the following concerns with this:
a. There can be presence of immune cells in the tumor core as well, so defining the non malignant area by their presence is a biased approach, and would result in a wrongly estimated near absence of immune cells in spots considered to be malignant.

Response: Thank you for the reviewer's comments. In this step, our purpose is to identify one cluster of spots, which clustered by morphologically adjusted gene expression matrix, as non-malignant area. The copy number variation (CNV) in this cluster with highest median NormalScore was defined as CNV reference, and we assess CNV level for remained spots by inferCNV. We did not assume absence of immune cells in spots considered to be malignant, and these spots could either be stroma region or other cell types region. We clarified this in the revised manuscript (page 26, line 642 - 643).

b. Even is this approach was valid, the list of genes is surprising: CCR7 can be expressed in most immune cells, but also by some tumor cells. AICDA is only expressed by germinal center B cells. CXCR6 and FOXP3 are related to some T cell subsets. It is surprising to see them considered at the same level as pan-immune markers or pan-T cell markers. There are additionally no markers of myeloid cells, such as macrophages (the most abundant immune cells in many malignancy!), dendritic cells, monocytes, neutrophils, ...

Response: Our purpose is finding a cluster with most immune cell enrichment through comparing the score of immune-related genes across the clusters. As suggested by reviewer, we revised the immune genes with only the lineage markers of lymphocytes, including pan-immune markers (PTPRC), pan-T cell markers (CD2, CD3D, CD3E, CD3G, CD5, CD7)⁵ and B cell markers (CD79A, MS4A1, CD19)⁶. We observed consistent results for updated lymphoid-related genes and immune-related genes in original manuscript (**Fig. A6**). We updated the gene signature in revised Cottzam.

Fig. A6. Comparison of normal clusters with different immune-related genes. Violin and box plots showed the signature scores of immune (upper, lymphoid signature-old) and lymphoid (bottom, lymphoid signature-update) related genes in CRC (n = 3), BRCA (n = 2), HCC (n = 3), intrahepatic cholangiocarcinoma11 (ICC, n = 1), clear cell renal cell carcinoma (ccRCC, n = 1), and ovarian cancer (OV, n = 1). The cluster with the highest signature score (red box) in each sample was selected as normal cluster and marked with a red box.

c. Why label non malignant tissue based on immune signatures? Why not include genes related with normal stroma?

Response: Thanks for the reviewer’s comment. To capture transcripts from Visium Spatial Gene Expression slide, the tissue need be permeabilized to release mRNA from the cells and mRNA binds with spatially barcoded oligonucleotides present on the capture area. Different cell types have different permeabilization conditions. Fibroblasts are more difficult to permeabilize than tumor cells or immune cells. When other cells are completely permeabilized, the permeabilization of fibroblast is often incomplete, resulting in relatively small **nCount** and **nFeature** of fibroblast

cells in ST. Therefore, using it as a reference cluster for normal CNVs will be unstable. In this study, we label nonmalignant tissue based on immune signatures as a reference cluster for normal CNVs.

3) Cottrazm's annotation of tumor region needs to be adequately validated against what could be considered as an appropriate gold standard. I would consider pathologist's annotations of tumour core, invasive margin and adjacent tissue to be gold standard. The authors should consider collaborating with one or several pathologist to evaluate the agreement between Cottrazm and pathologists, paying attention to separate their datasets into training and validation.

Response: Thanks for the reviewer's constructive suggestion. We obtained a new spatial transcriptomics of Squamous Cell Carcinoma (SCC) performed by 10X Genomics platform and the H&E staining with tumor boundary annotated by pathologist. We also asked the professional pathologist from Ruijin Hospital, China to help us annotate the tumor boundary in representative H&E staining which used in our study, including breast cancers (BRCA), colorectal cancer (CRC), and hepatocellular carcinoma (HCC) (**Fig.A7a**). Then, we imported the outlined tumor boundary layer into R and converted into an unordered set of points (pixel coordinates), each point defining a position on the tumor boundary layer. Further, we calculated pairwise distances between spots of ST and tumor boundary outline by k-Nearest Neighbor algorithm (kNN, dbscan R package) to find the minimum distance from a spot to the closest outline (**Fig. A7b-d**). We consider the tumor boundary spot predicted by our tool Cottrazm is match to the location of pathologist-annotated tumor boundary outline when their distance is less than one spot size. Our result showed predicted tumor boundary spot match to pathologist-annotated tumor boundary outline range 91.30% in SCC to 99.73% in CRC (**Fig.A7e**), suggesting the great performance of our tool on the delineation of tumor boundary. We added **Fig. A1** as **Fig. 2e-h**, **Supplementary Fig. 1f**, and **Supplementary Fig. 2b-d** and the result of concordance between pathologist annotation and prediction of Cottrazm in revised manuscript (page 8-9 line 185-201)

Fig. A7. The concordance of pathologist annotation and prediction of Cottrazm. (a) HE stained images with pathologist annotated tumor boundary of squamous cell carcinoma (SCC), breast cancer (BRCA), colorectal cancer (CRC), and hepatocellular carcinoma (HCC). (b) Tissue slides were annotated by malignant spots (Mal, red), boundary spots (Bdy, blue), and non-malignant spots (nMal, orange), including SCC, BRCA, CRC, and HCC. (c) Tumor boundary annotated by pathologist (black) and boundary spots

annotated by Cottrazm (Blue). **(d)** Line segments of the shortest distance from boundary spots annotated by Cottrazm to the pathologist's boundary. **(e)** Bar plot showing the proportion of boundary spots which distance to pathologist's boundary is less than or equal to one spot.

4) Spot deconvolution should be validated against other techniques as well, such as immunohistochemistry, not only against deconvolution methods. There is a public dataset of Visium and CD3 (T cells) staining on invasive ductal carcinoma that the authors could consider using to compare their estimates with immunostaining results: https://support.10xgenomics.com/spatial-gene-expression/datasets/1.2.0/V1_Human_Invasive_Ductal_Carcinoma

Response: Thanks for the reviewer's comment. We compared CD3 staining of IDC tissue section (**Fig. A8a**), we obtained the processed image which highlight CD3 (cyan) staining from a previous study⁷. Then, we compared deconvolution results of Cottrazm predicted T with CD3 staining in IDC sample. We found the location of predicted T cells was highly overlapped with CD3 fluorescence as well as the proportion was higher at the spot where the intensity of CD3 fluorescence was stronger (**Fig. 8b-c**). In addition, we examined the expression of CD3-encoding genes of the IDC tissue and found consistency with the CD3 fluorescence (**Fig. A8d**). We added **Fig. A8** as **supplementary Fig. 4e-h** and added this result in the revised manuscript (page 12, line 285-293).

Fig. A8 Comparison of the immunostaining, gene expression, deconvolution result of T cells in breast cancer. **(a)** Four-channel immunofluorescent imaging of the tissue section (N= 1 tissue section, n= 4,727 spots), depicting intensity of DAPI (green), the fiducial frame (blue), and CD3 (yellow). The FITC filter (magenta) does not correspond to an antibody stain. **(b)** Two stains, CD3 (cyan) and DAPI (blue) were used

to highlight CD3 florescence. (c) Predicted proportion within each capture spot for T cells in IDC sample, Color indicates the percentage of cell type. (d) Spatial feature plots showed the expression of CD3D (left), CD3E (middle), CD3G (right) of IDC sample.

5) In the benchmark against other deconvolution approaches, some methods are missing, including Stereoscope (<https://doi.org/10.1038/s42003-020-01247-y>) and Cell2Location (<https://doi.org/10.1038/s41587-021-01139-4>).

Response: We added Stereoscope⁸ and Cell2Location⁹ to the benchmark analysis, and then found that Cell2Location and Stereoscope had lower accuracy and specificity than Cottrazm and a higher probability of F1 error (**Fig. A9a**). In terms of correlation, Cell2Location and Stereoscope have comparable performance to Cottrazm (**Fig. A9b-d**). We added **Fig. A9a-d** as **Fig. 3a-d** and added this result in the revised manuscript (page 9, line 209-210 and page 10, line 229-235).

6) The use of Spearman correlation seems inadequate. A linear relationship between truth and prediction is expected, so Pearson correlation would be a much more appropriate metric.

Response: As suggested by the reviewer, we used the Pearson correlation to confirm the concordance between prediction and true proportion of cellular composition (**Fig. A9e-f**). We added **Fig. A9e-f** as **Supplementary Fig. 3a-b** and added this result in the revised manuscript (page 10, line 231-235).

Fig. A9. Benchmarking Cottrazm's performance of deconvolution using simulated data. (a) Benchmarking of classification performance of Cottrazm and other eight deconvolution tools on simulated mixtures, including accuracy, F1 score, and specificity. (b) A benchmark of the ability to distinguish different cell types across different deconvolution tools. Spearman's correlation was performed to evaluate the correlation between the predicted proportions and the ground truth for each cell type. (c) Benchmark of deconvolution tools' consistency of cell type distribution between the predicted proportions and the ground truth for each spot. The box plot reflects the overall distribution of Spearman's correlation calculated in each spot for each method. (d) Proportion prediction performance of the different deconvolution tools on simulated mixtures by Jensen-Shannon Divergence (JSD). (e) A benchmark of the ability to distinguish different cell types across different deconvolution tools. Pearson's correlation was performed to evaluate the correlation between the predicted proportions and the ground truth for each cell type. (f) Benchmark of deconvolution tools' consistency of cell type distribution between the predicted proportions and the ground truth for each spot. The box plot reflects the overall distribution of Pearson's correlation calculated in each spot for each method.

7) To my understanding, the results presented in Fig. 3e-h are based on the same scRNA-seq dataset used for training (generation of the simulated data) and validation. This is biased, and 3f's "truth" should come from an independent validation dataset that is not used to generate the simulated Visium data.

Response: We obtained an additional independent CRC scRNA-seq from a previous study¹⁰ as training dataset for an independent validation. We integrated the sub-spot matrix for prediction and truth and found consistent cell type clustering results (**Fig. A10a-c**). Spearman's and Pearson's correlation based on the predicted results and the true average expression of each gene in each sub-spot cell type show concordance, that the Spearman's correlation coefficient (R_S) and Pearson's correlation coefficient (R_P) of *Cottrazm-SpatialRecon* prediction and true proportion ranged from 0.77 in endothelial cells to 0.86 in malignant cells and myeloid cells, 0.81 in T cells to 0.89 in myeloid cells, respectively (**Fig. A10d-e**). This suggests that the prediction of *Cottrazm* can reflect cell type-specific gene expression profiles at the sub-spot level with high fidelity and high-resolution. We added **Fig. A10a** as **Supplementary Fig. 3c**, and **Fig. A10b-e** as **Fig. 3e-h** and added this result in the revised manuscript (page 10, line 244-246; page 11, line 254-258).

Fig. A10. Benchmarking Cottrazm's performance of reconstruction using simulated data. (a) UMAP projection of cell types in prediction results (left) and truth (right) simulated data. (b-c) UMAP projections of cell type specific gene expression profiles (GEP) at sub-spot level by integrating the predicted proportions and the ground truth of simulated mixtures. (b) colored by cell types, (c) colored by the prediction (orange) and the truth (green). (d) Heatmap showing the concordance between cell type proportions measured by Cottrazm and the ground truth of simulated mixtures by Spearman's correlation (upper) and Pearson's correlation (bottom). (e) Scatter plots depicting concordance between cell type proportions measured by Cottrazm and the ground truth of simulated mixtures, including tumor cells, T cells, myeloid cells, fibroblast cells, endothelial cells, and B/Plasma cells.

8) The simulated dataset was limited to a maximum of 10 cells/spot. Where does this limitation

come from? In dense areas, such as TLS, the number of spots can often be much higher than 10, probably around ~ 30 cell/spot. The authors should consider simulating some spots with higher cell type densities.

Response: Indeed, the number of cells captured in a single spot is based on the tissue type, cell size, and section thickness; this is generally between 1-10 cells follow the 10x Genomics Visium platform. As suggested by the reviewer, we set the number of spots in simulated dataset between 10~30 cell/spot to validate the robustness of our tool. The results were found to be consistent with the simulation data of 1-10 cells per spot (**Fig. A11**).

Fig. A11. Benchmarking Cottrazm's performance of deconvolution using simulated data (max number of cells in TLS spot = 30). (a) Benchmarking of classification performance of Cottrazm and other eight deconvolution tools on simulated mixtures, including accuracy, F1 score, and specificity. (b) A benchmark of the ability to distinguish different cell types across different deconvolution tools. Spearman's correlation was performed to evaluate the correlation between the predicted proportions and the ground truth for each cell type. (c) Benchmark of deconvolution tools' consistency of cell type distribution between the predicted proportions and the ground truth for each spot. The box plot reflects the overall distribution of Spearman's correlation calculated in each spot for each method. (d) Proportion prediction performance of the different deconvolution tools on simulated mixtures by Jensen-Shannon Divergence (JSD).

Minor points:

- 1) Some of the appropriate literature in the field is not cited.
- The authors should consider discussing their findings in light of the reports of the impact of TGFb leading to immune-excluding tumors unresponsive to ICB (<https://doi.org/10.1038/nature25501>).
 - The deleterious impact of SPP1+ macrophages has also already been reported (see <https://doi.org/10.1038/s41571-022-00620-6> for a review).
 - Ref 53 (Cabrita et al.) was published alongside 2 other articles (<https://doi.org/10.1038/s41586-019-1906-8> and <https://doi.org/10.1038/s41586-019-1922-8>) that extend the results to other cancer types, and this would further the discussion.
 - CellChat is used, but not cited (<https://doi.org/10.1038/s41467-021-21246-9>). Please consider citing it so that the authors can get credit for their work.

Response: We have cited these studies and discussed in revised manuscript.

- 2) The authors could consider adding a third UMAP on Fig. 4c that would show the region of origin (Mal/Bdy/nMal).

Response: We have added the third UMAP to present the region of origin (Mal/Bdy/nMal, Fig. A12) in Fig. 4c.

Fig. A12. UMAP projections of sub-spots in three CRC ST dataset predicted by Cottrazm, each dot denotes one sub-spot; color represents cluster origin (left panel), patient donors (middle panel), and the region of origin (right panel).

- 3) How was the TCGA dataset separated between High and Low boundary score? Was it median cut-off? This needs to be specified.

Response: The TCGA dataset was stratified by median ieScore, we stated this in figure legend of Fig. 5j (page 24, line 607-608).

- 4) Cottrazm seems to be only for 10x Genomics Visium data, not for other spatial transcriptomics modalities (e.g. SlideSeq2, Nanostring GeoMx, ...). This needs to be made clear in the manuscript.

Response: Yes, our tool is currently only available for 10x Genomics Visium data, we clarified this in the revised manuscript (page 4, line 84-85)

5) The language in which Cottrazm is available (R?) should be specified in the text.

Response: The Cottrazm software package depends on R language. We claimed this in the “Code availability” section (page 36, line 916).

Reviewer #3 (Remarks to the Author): Expert in spatial transcriptomics, scRNA-seq, and bioinformatics

Xun and colleagues present Cottrazm, a bespoke method for identifying malignant, boundary and non-malignant spots in spatial transcriptomics data. This method works by performing InferCNV on spatially resolved spots to identify potentially malignant spots, then expanding around the border of such malignant spots to assign boundary spots. The authors demonstrate the utility of Cottrazm by applying it to samples across several cancer types, and identify a common pattern of expression of expression, likely driven by localisation of specific subtypes of macrophage and fibroblast, present in the border regions across tumours. I find the work is innovative and promising for further understanding the tumour microenvironment. The manuscript is clear and well-motivated, and the software is available online. I have the following comments regarding the manuscript:

Response: We appreciated that the reviewer considers our study is innovative and promising.

- It's unclear what motivates the various choices within the Cottrazm-SpatialDecon method, e.g. what motivates the choice of PAGE for the gene set enrichment analysis, and DWLS as opposed to say typical least squares regression?

Response: Thanks for the reviewer's comment. PAGE calculates the enrichment score of a set of signature genes based on the expression fold change of each gene, which was calculated by using the expression value in one spot versus the mean expression of all spots¹¹. Compared to other enrichment algorithm, PAGE detected a larger number of significantly altered gene sets, suggesting that PAGE is more sensitive. PAGE was able to detect significantly changed gene sets from transcription data irrespective of different platforms¹². In conclusion, PAGE is more sensitive and faster when used for enrichment of large set of signatures and suitable for various sequencing platforms.

As the described in a previous study¹³, in the typical least squares regression tend to be biased against cell types that either make up a small proportion of the total bulk cell population, or are characterized by lowly expressed genes. The deconvolution problem is represented as a system of linear equations: $Sx=t$, where S is an $n \times k$ gene signature matrix (n =number of genes, k =number of cell types), t is an $n \times 1$ vector representing the bulk RNA-seq data, and x is a $k \times 1$ vector containing the cell-type numbers. Since typically $n \gg k$, this is an over-determined equation with no exact solution. In the typical least square regression approach, the solution x minimizes the total squared absolute error. Therefore, the typical least squares regression will result in two undesirable consequences, including the estimation error for rare cell types is typically large and not all informative genes are effectively taken into account. DWLS introduces a dampening constant d , which defines the maximum value any weight can take on. DWLS can accurately estimate rare cell types and properly adjust the contribution of each gene. In addition, Li *et.al.* benchmarked spatial and single-cell transcriptomics integration methods for cell type deconvolution and showed DWLS have great performance on cell type deconvolution¹⁴.

- How relevant is the size of the spots for determining? e.g. if the spot data was a higher resolution

would you be extracting different boundary regions? seems to be the case given the parametrisation on neighbouring spots. As it stands it's unclear whether such method is applicable to a technology with different spot size/resolution as the boundary could be thinner or thicker depending on the technology (and not the biology). The authors should discuss the relative merits to determining boundary by the nearest neighbors versus an absolute distance (e.g. in microns). The point above could be tested by artificially combining spots to a lower resolution and examining the concordance of malignant/boundary/non-malignant predictions.

Response: To test the influence of different spot size/resolution on the boundary region, we constructed simulated spatial data of different spot sizes. The bin size represents the diameter/edge length of the space point. The predicted boundaries at different bin sizes correspond to the real boundaries of the simulated spatial data (**Fig. A12**). Bin = 1 and bin = 2 are the same size and twice the diameter of a spot on the 10X Genome Visium platform, respectively. When the spot size with bin = 1, the predicted tumor boundary will be thinner, but the overlap ratio of the predicted boundary spot of bin = 1 (**Fig. A12a**) is up to 95.80% with predicted boundary spot of bin = 2 (**Fig. A12b, e**). Indeed, Cottrazm predicted tumor boundary is composed of spatial spots, the thickness of predicted boundary is correlated with the size of spot size of different technology. In Cottrazm, the diameter of spatial spots and the distance of neighbored spots were relatively determined by the spot size on sequencing slide and the resolution of HE/IF image. Relative diameter can be applied to variable sequencing resolution while the absolute distance needs to be re-adjusted according to different resolution of image and spatial spot, which has certain limitations. With the advantage of ST technology, most ST datasets have high resolution and spot of tumor boundary contain few cells. We discussed this in the revised manuscript (page 19, line 474-482).

Fig. A12. Predicted tumor boundary of different spot size (a, c) Simulated spatial slides were annotated by malignant spots (Mal, red), boundary spots (Bdy, blue), and non-malignant spots (nMal, orange), including bin=1 (a) and bin=2 (c). (b, d) Spatial scatter pie plots representing the actual proportions of the seven cell types in simulated ST slides of bin=1 (b) and bin=2 (d), with projection of predicted tumor boundary (blue circle). (e) Dot plot showed the overlap of the prediction boundaries for different bin values.

- How applicable is InferCNV to non-single-cell data as used here (per spot)? Could this lead to artefacts in the InferCNV output due to input of multiple cells, possibly mixtures of malignant and non-malignant cells? Additionally, it may be that not all cancers result in such large copy-number

changes but rather have a large SNV burden (e.g. melanoma), this limitation should be discussed in the text.

Response: The spot-level spatial inferCNV has been widely used in spatial transcriptomics of prostate cancer¹⁵, colorectal adenocarcinoma¹⁶, hepatocellular carcinoma¹⁷, ovarian cancer¹⁸, non-small cell lung carcinoma¹⁹, and etc. For further validation, we randomly selected 100 malignant cells in the Lee's CRC cohort to simulate spatial malignant data, with 1-10 malignant cells randomly included in each spot and found that the CNV score of a spot was not affected by the number of malignant cells within that spot (**Fig. A13a**). Indeed, SNV burden is more pronounced in melanoma, but CNV can still distinguish malignant regions from Bdy and nMal regions in human squamous cell carcinoma (**Fig. A13b**). We discussed this in the revised manuscript (page 18, line 457-461).

Fig. A13. CNV Score of simulated spatial data and human squamous cell carcinoma. (a) Box and violin plot showing the CNV score of simulated spatial data generated from malignant cells of human CRC. (b) Boxplot showing the CNV score calculated by R package infercnv in three regions of human squamous cell carcinoma.

- What ground truth is present? e.g. Figure 2a appears like these are the outputs of Cottrazm, are there expert/pathologist annotations of these slides to refer to?

Response: In the Figure 2a is the tumor boundary predicted by Cottrazm. To validate the robustness of Cottrazm, we obtained a new spatial transcriptomics of Squamous Cell Carcinoma (SCC) performed by 10X Genomics platform and the H&E staining with tumor boundary annotated by pathologist. We also asked the professional pathologist from Ruijin Hospital, China to help us annotate the tumor boundary in representative H&E staining which used in our study, including breast cancers (BRCA), colorectal cancer (CRC), and hepatocellular carcinoma (HCC) (**Fig. A14a**). Then, we imported the outlined tumor boundary layer into R and converted into an unordered set of points (pixel coordinates), each point defining a position on the tumor boundary layer. Further, we calculated pairwise distances between spots of ST and tumor boundary outline by k-Nearest Neighbor algorithm (kNN, dbscan R package) to find the minimum distance from a spot to the closest outline (**Fig. A14b-d**). We consider the tumor boundary spot predicted by our tool Cottrazm is match to the location of pathologist-annotated tumor boundary outline when their distance is less than one spot size. Our result showed predicted tumor boundary spot match to pathologist-annotated tumor boundary outline range from 91.30% in SCC to 99.73% in CRC, suggesting the great performance of our tool on the delineation of tumor boundary. We added **Fig. A1** as **Fig. 2e-h**, **Supplementary Fig. 1f**, and **Supplementary Fig. 2b-d** and the result of concordance between pathologist annotation and prediction of Cottrazm in revised manuscript (page 8-9 line 185-201)

Fig. A14. The concordance of pathologist annotation and prediction of Cottrazm. (a) HE stained images with pathologist annotated tumor boundary of squamous cell carcinoma (SCC), breast cancer (BRCA), colorectal cancer (CRC), and hepatocellular carcinoma (HCC). **(b)** Tissue slides were annotated by malignant spots (Mal, red), boundary spots (Bdy, blue), and non-malignant spots (nMal, orange), including SCC, BRCA, CRC, and HCC. **(c)** Tumor boundary annotated by pathologist (black) and boundary spots annotated by Cottrazm (Blue). **(d)** Line segments of the shortest distance from boundary spots

annotated by Cottrazm to the pathologist's boundary. (e) Bar plot showing the proportion of boundary spots which distance to pathologist's boundary is less than or equal to one spot.

- What is the false negative rate of Cottrazm? i.e. will it always assign spots to "Malignant"? This can be tested by performing Cottrazm on an adjacent normal tissue.

Response: The reviewer is correct. If the slide has only the normal tissue, Cottrazm may predict highest CNV score cluster as malignant core. Therefore, we should only apply Cottrazm to the tumor tissue slides. We clarified this in the revised manuscript (page 2, line 24).

- The manuscript states "All scripts used to reproduce all the analyses are also available at the same website." but I can't seem to find this on the Github page. Please specifically link to the scripts and/or make these available on the Github.

Response: We have provided package vignette to reproduce the process of Cottrazm in the GitHub. And scripts used to reproduce other analyses are available from the author upon reasonable request. We re-clarified this in the "Code availability" section (page 36, line 916-920).

- No plots visible in the package vignette as all code is commented out, this should be amended to display the expected output in the vignette.

Response: We updated vignette to display the output in the vignette.

Minor

- Figure 1 schematic of tabular representation of "Spots" and "HE stain" doesn't quite make sense
- the dimensionality is very unlikely to be equal to the genes in the "Spots" "Feature" matrix, and it's unclear what the rows actually represent, the HE stain looks like both rows and columns are spots and the colour of the cells themselves are the HE stain values. In addition, it would be useful to include a colour legend for this schematic too, what do the light to dark pink colours represent in the "Spots" "Morph adjusted feature" matrix?

Response: We changed the "HE stain" to "Pixel" in the revised Fig 1, which means the HE stained image was segmented into a tile file containing pixel spots and extract the features of each pixel spot to form a morphological matrix.

- typo in Figure 1 "Neighbors" and "Figrue"

Response: We corrected them.

- Figure 2 b boxes overlaid on violin plots should either be reordered or some transparency used, as it masks the violin's distribution

Response: We used 50% transparency in the revised Fig. 2b boxes.

Reference

1. Ji, A. L. *et al.* Multimodal Analysis of Composition and Spatial Architecture in Human Squamous Cell Carcinoma. *Journal of Cleaner Production* **182**, (2020).
2. Uhlén, M. *et al.* Tissue-based map of the human proteome. *Science* **347**, (2015).
3. Zhang, L. *et al.* Single-Cell Analyses Inform Mechanisms of Myeloid-Targeted Therapies in Colon Cancer. *Cell* **181**, 442–459.e29 (2020).
4. Cheng, S. *et al.* A pan-cancer single-cell transcriptional atlas of tumor infiltrating myeloid cells. *Cell* **184**, 792–809.e23 (2021).
5. Darzynkiewicz, Z., Roederer, M. & Tanke, H. J. Cytometry : New Developments.
6. Kaminski, D. A., Wei, C., Qian, Y., Rosenberg, A. F. & Sanz, I. Advances in human B cell phenotypic profiling. *Frontiers in Immunology* **3**, 302 (2012).
7. Zhao, E. *et al.* Spatial transcriptomics at subspot resolution with BayesSpace. *Nature Biotechnology* (2021). doi:10.1038/s41587-021-00935-2
8. Andersson, A. *et al.* Single-cell and spatial transcriptomics enables probabilistic inference of cell type topography. *Communications Biology* **3**, 1–8 (2020).
9. Kleshchevnikov, V. *et al.* Cell2location maps fine-grained cell types in spatial transcriptomics. *Nature Biotechnology* (2022). doi:10.1038/s41587-021-01139-4
10. Lee, H. O. *et al.* Lineage-dependent gene expression programs influence the immune landscape of colorectal cancer. *Nature Genetics* **52**, 594–603 (2020).
11. Dries, R. *et al.* Giotto: a toolbox for integrative analysis and visualization of spatial expression data. *Genome Biology* **22**, 1–31 (2021).
12. Kim, S. Y. & Volsky, D. J. PAGE: Parametric analysis of gene set enrichment. *BMC Bioinformatics* **6**, (2005).
13. Tsoucas, D. *et al.* Accurate estimation of cell-type composition from gene expression data. *Nature Communications* **10**, (2019).
14. Li, B. *et al.* Benchmarking spatial and single-cell transcriptomics integration methods for transcript distribution prediction and cell type deconvolution. *Nature Methods* (2022). doi:10.1038/s41592-022-01480-9
15. Erickson, A. *et al.* Spatially resolved clonal copy number alterations in benign and malignant tissue. *Nature* **608**, (2022).
16. Zhang, R. *et al.* Spatial transcriptome unveils a discontinuous inflammatory pattern in proficient mismatch repair colorectal adenocarcinoma. *Fundamental Research* 1–8 (2022). doi:10.1016/j.fmre.2022.01.036
17. Wu, R. *et al.* Comprehensive analysis of spatial architecture in primary liver cancer. *Science Advances* **7**, (2021).
18. Kock, L. De *et al.* Spatial transcriptomics reveals ovarian cancer subclones with distinct tumour microenvironments. (2022).
19. Zhang, L. *et al.* Spatial transcriptome sequencing revealed spatial trajectory in the Non-

Small Cell Lung Carcinoma. *bioRxiv* 2021.04.26.441394 (2021).
doi:10.1101/2021.04.26.441394

REVIEWER COMMENTS

Reviewer #2 (Remarks to the Author):

This revised version, the authors have put a lot of work and adequately addressed most of my comments, and I think the manuscript has greatly improved. I still have slight remaining concerns as follow (the numbers refer the the original points I had on the original submission):

2) While it now clear how Mal spots are being selected using inferCNV, I am still not convinced by the list of immune genes adopted by the authors. Have only genes related to T cells and B cells is a bit biased. In samples having a lot of tertiary lymphoid structures, this would highlight only these TLS and not the overall nMal spots. A simple fix would be to incorporate also genes for cell types that are more common outside of TLS than inside of them, notably macrophages.

4) I appreciate the effort made to compare the output of deconvolution with IHC data. However, this should be quantified and not simply be put visually as is currently the case. Maybe be looking at the correlation between staining intensity and predicted cell proportion within each spot? Or any other quantitative way to compare both results.

All other points have been convincingly addressed, and I congratulate the authors for this good revision.

Reviewer #3 (Remarks to the Author):

I thank the authors for their careful work in addressing my and other reviewers' comments. In particular I appreciate the incorporation of pathologists' annotation as a ground truth for comparison with cottrazm, alongside the additional simulations demonstrating generalisability of the method to other spatial omics technologies with different resolutions.

I would appreciate if the motivation of PAGE and DWLS methods that are used within cottrazm are incorporated into the manuscript itself. This will make it clear for the research community why such choices are made, leading to higher take-up for analyses.

Point-by point Responses to Reviewers' Comments:

Reviewer #2 (Remarks to the Author):

This revised version, the authors have put a lot of work and adequately addressed most of my comments, and I think the manuscript has greatly improved. I still have slight remaining concerns as follow (the numbers refer the the original points I had on the original submission):

Response: We thank the reviewer for positive comments on our revised manuscript.

2) While it now clear how Mal spots are being selected using inferCNV, I am still not convinced by the list of immune genes adopted by the authors. Have only genes related to T cells and B cells is a bit biased. In samples having a lot of tertiary lymphoid structures, this would highlight only these TLS and not the overall nMal spots. A simple fix would be to incorporate also genes for cell types that are more common outside of TLS than inside of them, notably macrophages.

Response: We thank the reviewer's suggestion. We added pan-macrophage markers (CD14, FCGR3A, CD68, FCGR1A)¹ into the original lymphoid signatures. We observed the consistent result for signatures with macrophage markers, but the signature score difference decreased among clusters (**Fig. A1**). Our purpose is to find a cluster with most lymphoid cell enrichment as the reference cluster in *InferCNV::run* function but not the overall nMal spots. Therefore, we used the lymphoid-related genes in Cottrazm.

Fig. A1. Comparison of normal clusters with different immune-related genes. Violin and box plots showed the signature scores of lymphoid signatures (upper) and signatures with macrophage markers (bottom) in CRC (n = 3), BRCA (n = 2), HCC (n = 3), intrahepatic cholangiocarcinoma (ICC, n = 1), clear cell renal cell carcinoma (ccRCC, n = 1), and ovarian cancer (OV, n = 1).

4) I appreciate the effort made to compare the output of deconvolution with IHC data. However, this should be quantified and not simply be put visually as is currently the case. Maybe be looking at the correlation between staining intensity and predicted cell proportion within each spot? Or any other quantitative way to compare both results.

Response: Thanks for the reviewer's comment. We downloaded the grayscale image of anti-CD3 antibody single channel from 10X Genomics website (**Fig. A2a**) and obtained the morphology matrix with CD3 signal of each spatial spot. Then we normalized the signal by total intensity to obtain the CD3 intensity score. We projected CD3 intensity score and the proportion of T cell infiltration in spatial and they have consistent spatial distribution (**Fig. A2b-c**), and the significantly positive correlation between CD3 intensity score and proportion of T cell in each spot (**Fig. A2d**). We added

Fig. A2 as Supplementary Fig. 4e-h and added this result in the revised manuscript (page 12, line 298-299).

Fig. A8 The correlation between CD3 immunofluorescent intensity and T cell infiltration in breast cancer. (a) Anti-CD3 antibody one-channel immunofluorescent imaging of the tissue section (N = 1 tissue section, n = 4,727 spots). (b-c) Spatial feature plots showed the CD3 intensity score (b) and the predicted proportion within each capture spot for T cells (c), Color indicates the percentage of cell type. (d) Scatter plot depicting the correlation between CD3 intensity score and predicted proportion of T cell infiltration.

All other points have been convincingly addressed, and I congratulate the authors for this good revision.

Response: We are very pleased that the reviewer find the revisions adequate.

Reviewer #3 (Remarks to the Author):

I thank the authors for their careful work in addressing my and other reviewers' comments. In particular I appreciate the incorporation of pathologists' annotation as a ground truth for comparison with cottrazm, alongside the additional simulations demonstrating generalisability of the method to other spatial omics technologies with different resolutions.

Response: We are very pleased that the reviewer stratified our revised work.

I would appreciate if the motivation of PAGE and DWLS methods that are used within cottrazm are incorporated into the manuscript itself. This will make it clear for the research community why such choices are made, leading to higher take-up for analyses.

Response: We added the motivation of PAGE and DWLS methods in revised manuscript (page 7, line 147-149, line 153-155)

Reference

1. Murray, P. J. & Wynn, T. A. Protective and pathogenic functions of macrophage subsets. *Nat Rev Immunol* **11**, 723–737 (2011).